# TeLoGraF: Temporal Logic Planning via Graph-encoded Flow Matching

Yue Meng [1]   Chuchu Fan [1]

## Abstract

Learning to solve complex tasks with signal temporal logic (STL) specifications is crucial to many real-world applications. However, most previous works only consider fixed or parametrized STL specifications due to the lack of a diverse STL dataset and encoders to effectively extract temporal logic information for downstream tasks. In this paper, we propose TeLoGraF, Temporal Logic Graph-encoded Flow, which utilizes Graph Neural Networks (GNN) encoder and flow-matching to learn solutions for general STL specifications. We identify four commonly used STL templates and collect a total of 200K specifications with paired demonstrations. We conduct extensive experiments in five simulation environments ranging from simple dynamical models in the 2D space to high-dimensional 7DoF Franka Panda robot arm and Ant quadruped navigation. Results show that our method outperforms other baselines in the STL satisfaction rate. Compared to classical STL planning algorithms, our approach is 10-100X faster in inference and can work on any system dynamics. Besides, we show our graph-encoding method's capability to solve complex STLs and robustness to out-distribution STL specifications. Code is available at https://github.com/mengyuest/TeLoGraF.

## 1. Introduction

Learning to plan for complex tasks with temporal dependency and logical constraints is critical to many real-world applications, such as navigation, autonomous vehicles, and industrial assembly lines. For example, a robot might need to reach one of the destination regions in 10 seconds while avoiding obstacles, a robot arm will need to pick the objects in a specific sequence, a vehicle should come to a complete stop before the stop sign and may proceed only if no other cars have the right of way, etc. These cases require precise reasoning and planning to ensure safety, efficiency, and correctness. Hence, it is of utmost importance to endow agents with the ability to tackle temporal logic constraints.

Existing temporal logic specifications can be mainly categorized into linear temporal logic (LTL) (Pnueli, 1977), computation tree logic (CTL) (Clarke & Emerson, 1981), metric temporal logic (MTL) (Koymans, 1990), and signal temporal logic (STL) (Donzé & Maler, 2010; Raman et al., 2015; He et al., 2022). LTL checks a single system trace via logical operators and temporal operators, whereas CTL reasons for a tree of possible futures. Extending from LTL, MTL introduces timed intervals for the temporal operators and STL further generalizes this by handling continuous-valued signals. In this work, we focus on STL since it offers the most expressive framework to handle a wide range of requirements in robotics and cyber-physical systems.

However, it is challenging to plan paths under STL specifications, due to its non-Markovian nature and lack of efficient decomposition. Synthesis for STL satisfaction is NP-hard (Kurtz & Lin, 2022). Classical methods for solving STL include sampling-based methods (Vasile et al., 2017; Kapoor et al., 2020), optimization-based methods (Sadraddini & Belta, 2015; Kurtz & Lin, 2022; Sun et al., 2022) and gradient-based methods (Dawson & Fan, 2022) but they cannot balance solution quality and computation efficiency for high-dimensional systems or complex STLs.

There is a growing trend to use learning-based methods to satisfy a given STL (Li et al., 2017; Puranic et al., 2021; Liu et al., 2021; Guo et al., 2024) or a family of parametrized STL (Leung & Pavone, 2022; Meng & Fan, 2024), but few of them can train one model to handle general STLs. If a new STL comes, they must retrain the neural network to learn the corresponding policy, resulting in inefficiency. Hashimoto et al. (2022) propose a model-based approach to handle flexible STL forms but requires differentiable environments. While other works (Zhong et al., 2023; Feng et al., 2024) explored regularizing the pre-trained models with temporal logic guidance at test time, the generated trajectories are heavily constrained by the original data distribution. To the best of our knowledge, there is no existing model that can take a general STL as input to produce satisfiable solutions.

---

[1]Department of Aeronautics and Astronautics, MIT, Cambridge, USA. Correspondence to: Yue Meng <mengyue@mit.edu>.

*Proceedings of the 42$^{nd}$ International Conference on Machine Learning*, Vancouver, Canada. PMLR 267, 2025. Copyright 2025 by the author(s).

Impeding the advance of STL-conditioned models are three critical challenges: (1) most of the papers either work on simplified STLs that are not diverse enough or useful but heavily engineer-designed STLs (Maierhofer et al., 2022; Meng & Fan, 2023) that are hard to generalize (2) there is no large-scale dataset available to provide paired demonstrations with diverse STL specifications and (3) unlike visual-conditioned or language-conditioned tasks, there lacks an analysis of the effective encoder design to embed the STL information to the downstream neural network.

In this paper, we tackle all these points above and propose TeLoGraF (Temporal Logic Graph-encoded Flow), a graph-encoded flow matching model that can handle general STL syntax and produce satisfiable trajectories. We first identify four commonly used STL templates and collect over 200K diverse STL specifications. We obtain paired demonstrations using off-the-shelf solvers under each robot domain. Finally, we argue that GNN is a suitable encoder to embed STL information for the downstream tasks and we systematically compare different encoder architectures for STL encoding over varied tasks.

Extensive experiments have been conducted over five simulation environments, ranging from simple linear and Dubins car dynamics in the 2D space, to high-dimensional Franka Panda robot arm and Ant quadruped maze navigation tasks. Compared to other encoder architectures, our GNN-based encoder can produce the highest quality solutions. Our method also outperforms other guidance-based imitation learning methods such as CTG (Zhong et al., 2023) and LTL-DoG (Feng et al., 2024), demonstrating the need to bring STL into the training phase. Compared to classical methods (gradient-based (Dawson & Fan, 2022), CEM (Kapoor et al., 2020)), our approach is faster in inference and can work on any system dynamics and STL formats. Additional results also show our methods' capability in handling complex STLs and out-distribution specifications.

Our contributions are as follows: (1) we are the first to learn a generative model for planning tasks conditioned on diverse STL specifications (2) we identify four key STL patterns and collect over 200K diverse STLs paired with demonstrations in five simulation environments (3) our proposed TeLoGraF demonstrates strong performance over other baselines and encoder architectures, which supports the design of using graph-encoding to extract different STLs (4) all the code and the datasets will be open-sourced to promote the development of STL planning.

## 2. Related work

**Generative models for robotics.** The rise of diffusion models in computer vision (Ho et al., 2020) has drawn massive attention in the robotic community. We refer the readers

to this survey (Urain et al., 2024) for a full review. The general paradigm is to collect demonstrations from real-world (Chi et al., 2023; Zhong et al., 2023) or off-the-shelf solvers (Yang et al., 2023; Carvalho et al., 2023; Huang et al., 2024), and then train the neural network to imitate the data. Based on the input modality, the related works can be summarized into state-based (Janner et al., 2022), 2D diffusion policy (Chi et al., 2023), 3D diffusion policy (Ze et al., 2024; Ke et al., 2024), and vision language action models (Team et al., 2024). Varied output forms have been proposed, such as configuration-based (Yang et al., 2023), action-based (Chi et al., 2023; Ze et al., 2024), trajectory-based (Carvalho et al., 2023; Meng & Fan, 2024), and hierarchical models (Li et al., 2023; Chen et al., 2024). Regarding long-horizon task planning, compositional diffusion has been studied in GSC (Mishra et al., 2023) ChainedDiffuser (Xian et al., 2023) and DiMSam (Fang et al., 2024), where the task skeletons are usually fixed. More recently, flow-based methods (Chang et al., 2022; Chen et al., 2023; Prasad et al., 2024; Zhang et al., 2024) appear to have more stable training and faster sampling speed than diffusion models. Hence, we decide to use the linear flow discussed in (Liu et al., 2022) as our generative model backbone.

**Signal Temporal Logic.** Decades of efforts have been devoted to controlling robots under temporal and logical constraints (Fainekos et al., 2009; Wongpiromsarn et al., 2012). Unlike linear temporal logic (Finucane et al., 2010; Vaezipoor et al., 2021), STL is not equipped with a well-formed automaton or progression mechanism (Bacchus & Kabanza, 2000) hence special treatment is needed for planning. Previous works include sampling-based method (Vasile et al., 2017), mixed-integer programming (MILP) (Sadraddini & Belta, 2015; Sun et al., 2022), evolutionary algorithms (Kapoor et al., 2020), gradient-based method (Dawson & Fan, 2022; Leung et al., 2023), reinforcement learning (Li et al., 2017) and model-based learning (Liu et al., 2021; Meng & Fan, 2023; Eappen et al., 2025). Temporal logic has been used to guide the diffusion models sampling in CTG (Zhong et al., 2023) for STL and in LTLDoG (Feng et al., 2024) for LTL, but they do not incorporate STL syntax during the training. Meng & Fan (2024) handles diverse STL in training but is limited to continuous parameters, whereas ours emphasizes handling general STL structures. The most similar to ours is Hashimoto et al. (2022), which studies different STL encoder architectures for model-based learning, but they work on shorter and simpler specifications without obstacles.

## 3. Preliminaries

### 3.1. Signal Temporal Logic (STL)

Consider a discrete-time system $x_{t+1} = f(x_t, u_t)$ where $x_t \in \mathbb{R}^n$ is the state, and $u_t \in \mathbb{R}^m$ is the control. Start-

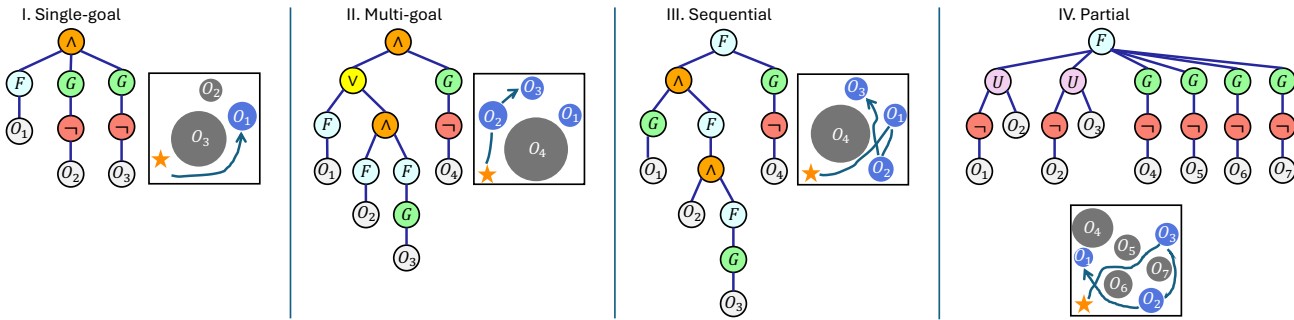

Figure 1: Four STL templates used in this paper. **Single-goal**: reach one goal under time constraints while avoiding obstacles. **Multi-goal**: reach one of the valid subsets of the goals. **Sequential**: All the goals needed to be reached in a strict temporal order. **Partial**: Some goals must be reached first before reaching other goals (no global order is explicitly specified).

ing from an initial state $x_0$, a signal $s = x_0, x_1, ..., x_T$ is generated via controls $u_0, ..., u_{T-1}$. STL specifies signal properties via the following rules (Donzé et al., 2013):

$$\phi ::= \top \mid \mu(x) \geq 0 \mid \neg\phi \mid \phi_1 \wedge \phi_2 \mid \phi_1 U_{[a,b]} \phi_2. \quad (1)$$

Here the boolean-type operators split by "|" are the building blocks to compose an STL: $\top$ means "true", $\mu$ denotes a function $\mathbb{R}^n \to \mathbb{R}$, and $\neg, \wedge, U, [a,b]$ are "not", "and", "until" and the time interval from $a$ to $b$. Other operators are "or": $\phi_1 \vee \phi_2 = \neg(\neg\phi_1 \wedge \neg\phi_2)$, "eventually": $\Diamond_{[a,b]}\phi = \top U_{[a,b]}\phi$ and "always": $\Box_{[a,b]}\phi = \neg\Diamond_{[a,b]}\neg\phi$. We denote $s, t \models \phi$ if the signal $s$ from time $t$ satisfies the STL formula (the evaluation of $\phi$ returns True). For operators $\top, \mu > 0, \neg, \wedge$ and $\vee$, the evaluation checks for the signal state at time $t$. As for temporal operators (Maler & Nickovic, 2004): $s, t \models \Diamond_{[a,b]}\phi \Leftrightarrow \exists t' \in [t+a, t+b] \ s, t' \models \phi$; and $s, t \models \Box_{[a,b]}\phi \Leftrightarrow \forall t' \in [t+a, t+b] \ s, t' \models \phi$; and $s, t \models \phi_1 U_{[a,b]}\phi_2 \Leftrightarrow \exists t' \in [t+a, t+b], s, t' \models \phi_2, \forall t'' \in [0, t'], x, t'' \models \phi_1$. In plain words, $\phi_1 U_{[a,b]}\phi_2$ means "always hold $\phi_1$ true until $\phi_2$ happens in $[a, b]$." Robustness score (Donzé & Maler, 2010) $\rho(s, t, \phi)$ measures how well a signal $s$ satisfies $\phi$, where $\rho \geq 0$ iff $s, t \models \phi$. The score is computed as:

$$\rho(s, t, \top) = 1, \quad \rho(s, t, \mu) = \mu(s(t))$$
$$\rho(s, t, \neg\phi) = -\rho(s, t, \phi)$$
$$\rho(s, t, \phi_1 \wedge \phi_2) = \min\{\rho(s, t, \phi_1), \rho(s, t, \phi_2)\}$$
$$\rho(s, t, \Diamond_{[a,b]}\phi) = \sup_{r \in [a,b]} \rho(s, t+r, \phi)$$
$$\rho(s, t, \Box_{[a,b]}\phi) = \inf_{r \in [a,b]} \rho(s, t+r, \phi)$$
$$\rho(s, t, \phi_1 U_{[a,b]}\phi_2) =$$
$$\sup_{t' \in [t+a, t+b]} \min\left\{\rho(s, t', \phi_2), \inf_{t'' \in [t, t']} \rho(s, t'', \phi_1)\right\}$$
$$(2)$$

### 3.2. Diffusion models and flow matching

Diffusion models and flow matching learn a distribution from data samples. They both contain a forward process (for training) and an inverse process (for generating samples). For DDPM diffusion models (Ho et al., 2020), the data are diffused with Gaussian noise iteratively towards a white noise, and a neural network is trained to predict the noise at different diffusion steps. In the denoising process (inverse process), the samples initialized from the white noise are recovered gradually by removing the "noise" predicted by this neural net. For flow matching (Liu et al., 2022), in the forward process, the samples are continuously transformed into a Gaussian noise by a vector field, and the neural network learns to predict the vector field. During the inverse process, new samples are generated by integrating the predicted vector field over over time, starting from noise and progressively refining the samples.

## 4. Methodology

### 4.1. Problem formulation

Given a set of demonstrations $\mathcal{D} = \{(\phi_i, \{\tau_{i,k}\}_{k=1}^K)\}_{i=1}^N$ where $\phi_i$ is the STL formula defined in Sec. 3.1, and $\{\tau_{i,k}\}_{k=1}^K$ is a batch of trajectories that satisfy the STL specification, $\forall k, \tau_{i,k}, 0 \models \phi_i$, our goal is to learn a conditional generative model $p_\theta(\tau | x_0, \phi)$, so that given a query STL specification $\phi$ and an initial state $x_0$, the trajectories sampled from this learned distribution $\tau \sim p_\theta(\cdot | \phi, x_0)$ can satisfy the STL specification, i.e., $\tau, 0 \models \phi$.

### 4.2. STL specification templates

We aim to collect specifications that are both general and representative of the unique characteristics of the STL. Since our focus is primarily on addressing the complexity of STL rather than the skills required to perform each subtask, we restrict the set of atomic propositions semantics to "reach"

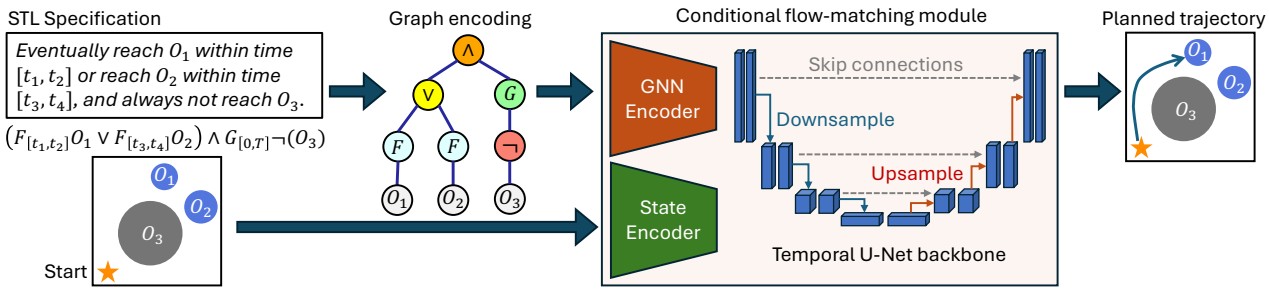

Figure 2: System diagram for TeLoGraF. The STL specification is encoded as a graph structure data. The GNN extracted embedding and the state embedding are sent to the Temporal U-Net flow model to generate the planned trajectory (via ODE).

(an object / a region) and "avoid" (an object / a region), while excluding operations that involve dexterity or agility skills. We want to emphasize the "temporal" and the "logical" facets of the STL specifications, where the former includes both absolute time constraints or relative temporal ordering (dependencies), and the latter captures patterns to describe multi-choice and selections. In this paper, we consider four types of STL templates: single-goal, multi-goal, sequential, and partial-order, denoted as $\phi_{single}$, $\phi_{multi}$, $\phi_{sequential}$, and $\phi_{partial}$ in the following Backus-Naur Form (BNF) (similar to Sec. 3.1, the symbols on the left-hand side of "::=" can take any of the forms split by "|" shown on the right-hand side of the expression):

$$\phi_{reach} ::= F_{[t_a,t_b]}O \mid F_{[t_a,t_b]}G_{[t_c,t_d]}O$$
$$\phi_{avoid} ::= \top \mid G_{[0,T]}\neg O \mid \phi_{avoid,1} \wedge \phi_{avoid,2}$$
$$\phi_{and} ::= \bigwedge_{i=1}^{n_1} \phi_{reach,i} \bigwedge_{j=1}^{n_2} (\phi_{or,j})$$
$$\phi_{or} ::= \bigvee_{i=1}^{n_1} \phi_{reach,i} \bigvee_{j=1}^{n_2} (\phi_{and,j})$$
$$\phi_{single} ::= \phi_{reach} \wedge \phi_{avoid}$$
$$\phi_{multi} ::= \phi_{and} \wedge \phi_{avoid} \mid \phi_{or}$$
$$\phi_{sequential} ::= \phi_{reach} \mid F_{[t_a,t_b]}(O \wedge \phi_{sequential}) \wedge \phi_{avoid}$$
$$\phi_{partial} ::= \bigwedge_{i=1}^{n_1} \neg O_{p_i} U_{[0,T]} O_{q_i} \wedge \bigwedge_{j=1}^{n_2} \phi_{reach,r_j} \wedge \phi_{avoid}$$

(3)

where $O_i$ is the atomic proposition representing "reach the $i$-th object", and $\neg O$ means "avoid the object". $\phi_{reach}$ means "eventually reach the object at time interval $[t_a, t_b]$ (and stay there for the time interval $[t_a + t_c, t_b + t_d]$ if $G_{[t_c,t_d]}$ is specified)". $\phi_{avoid}$ means "always avoid obstacles for all time (here $[0,T]$ denotes the full time horizon)". $\phi_{and}$ and $\phi_{or}$ mean "reach a subset of objects" (for example, $\phi_{reach,1} \vee (\phi_{reach,2} \wedge \phi_{reach,3})$ means "reach $O_1$ or reach both $O_2$ and $O_3$").

A graphical illustration for the examples can be found in

Figure. 1. The template $\phi_{single}$ specifies the single-goal-reaching with time constraints. The template $\phi_{multi}$ specifies a subset of goals to reach. The third template $\phi_{sequential}$ specifies multiple goals that must be reached in a specific order. And the last template $\phi_{partial}$ specifies some objects ($O_{p_i}$) need to be reached after other objects ($O_{q_i}$) are reached, and some objects needed to be reached within time constraints ($O_{r_j}$). All the templates are further paired with $\phi_{avoid}$ to reflect safety constraints.

### 4.3. Encoder design

To capture the information from an STL, we treat the STL as a syntax tree, and create a graph node for every operator and atomic proposition node in the syntax tree. The graph nodes are connected based on the edges on the syntax tree, where for every connection, we use a directed edge pointing from the child node to its father node. Mathmatically, given an STL syntax tree, we represent it as a directed graph[1] $G = (V, E, H) \in \mathcal{G}$ with nodes set $V$, edge set $E$ and node features $H$ (edges pointing from the child node to its father node), where each node feature $h_v \in \mathbb{R}^{d_G}$ contains all the attributes to characterize an STL operator or atomic proposition (AP), including the operator type $I_{type} \in \mathbb{Z}$, start time and end time $t_{start}, t_{end} \in 0, ..., T$ (for operators and APs that do not have the time intervals, we use -1 as the default value), object coordinates $x, y, z$ and radius $r$ (or side length, depending on whether the object is a circle/sphere or a square/cube). Besides, for the Until operator, we need to distinguish its left child with its right child as the order matters, so we use a separate binary variable to indicate whether it is the left child of the Until operator. In total, the node feature dimension is $d_G = 8$.

A multi-layer GNN $\mathcal{F}_\theta : \mathcal{G} \to \mathbb{R}^{d_z}$ operates on this graph data $G$ to get a $d_z$-dimension embedding. It first iteratively updates the node representations through message passing. The initial node feature for the node $v$ is $h_v^{(0)} = h_v$ dis-

---

[1]Here, we overload the symbol $G$ previously used as the "always" operator for consistency with the normal notation.

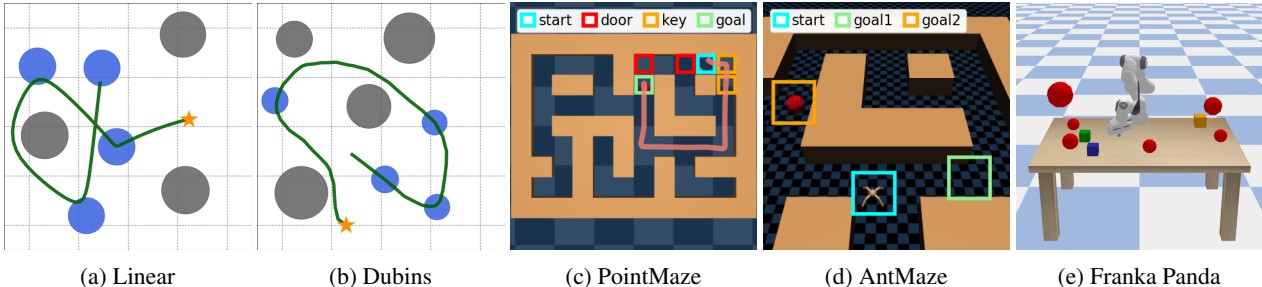

| (a) Linear | (b) Dubins | (c) PointMaze | (d) AntMaze | (e) Franka Panda |

Figure 3: Simulation benchmarks. In **Linear** and **Dubins**, a moving robot needs to reach circular regions and avoid circular obstacles. In **PointMaze** and **AntMaze**, the agent needs to reach/avoid square tiles in a maze. In **Franka Panda**, the robot arm needs to reach certain cubes on the table while not colliding with red balls.

cussed above. At each layer $l$ of the GNN, the node $v$'s feature $h_v^{(l)}$ is updated based on the message passing procedure with its neighbors (children) $\mathcal{N}(v)$:

$$h_v^{(l+1)} = \gamma\left(h_v^{(l)}, \bigoplus_{u \in \mathcal{N}(v)} \psi(h_v^{(l)}, h_u^{(l)})\right) \quad (4)$$

here $\gamma(.)$ and $\psi$ are the nonlinear update and message functions to increase the expressiveness of the GNN, and $\oplus$ is the permutation-invariant function (e.g., sum, mean, min, max). After $L$ layers of message passing, the final representation $h_v^{(L)} \in \mathbb{R}^{d_z}$ is read out using another aggregation function $h_G = \bigoplus_{v \in V} h_v^{(L)}$, which will be used for the downstream trajectory generation task.

### 4.4. Expressiveness of GNN to encode STL

It is crucial to verify theoretically that the GNN can distinguish different STLs. The work (Xu et al., 2018) shows GNN has (at most) the same expressiveness as the 1-dimensional Weisfeiler Leman graph isomorphism test (WL-test). It has been shown in (Kiefer, 2020) that the WL-test is a complete test for trees, hence GNN can distinguish two STLs (syntax trees) if they are different on the graph level.

### 4.5. Conditional flow for trajectory generation

Given the STL embedding $z \in \mathbb{R}^{d_z}$ for $\phi$ and the initial state $x_0 \in \mathbb{R}^n$, we aim to learn a conditional flow neural network $\mathcal{H}_\omega : \mathbb{R}^n \times \mathbb{R}^{d_z} \times \mathbb{R}^1 \times \mathbb{R}^{T \times (m+n)} \to \mathbb{R}^{T \times (m+n)}$ to generate the trajectory $\hat{\tau} \in \mathbb{R}^{T \times (m+n)}$ that can satisfy the STL. In each training step, we randomly draw a demonstration[2] $X_1 = \tau$ with its corresponding STL $\phi$ from the dataset ($x_0$ is the first state in $\tau$) and randomly draw $X_0$ from the Gaussian distribution $\mathcal{N}(0, I)$ and denote $\Delta X = X_1 - X_0$. Then, we uniformly sample $t \sim \text{Uniform}(0, 1)$ and take the linear combination: $X_t = tX_1 + (1 - t)X_0$. We denote $G$

---

[2]We use $X_0$ to denote the samples from the source distribution, and $X_1$ for the samples from the target data distribution, while using $x$ for the states in the trajectory.

as the graph for $\phi$ as mentioned in Sec. 4.3. To this end, we can formulate the Flow Matching loss:

$$\mathcal{L}_{FM} = \mathbb{E}_{t,X_0,X_1}||\mathcal{H}_\omega\left(\mathcal{F}_\theta(G), x_0, t, X_t\right) - \Delta X||^2 \quad (5)$$

which is used to update the neural network parameters $\theta$ and $\omega$ in the training. The network $\mathcal{H}_\omega$ learns a velocity field (conditional on $\phi$ and $x_0$) which determines a flow $\psi : [0, 1] \times \mathbb{R}^{T \times (m+n)} \to \mathbb{R}^{T \times (m+n)}$, depicted as $\frac{d}{dt}\psi(t, X) = \mathcal{H}_\omega(\mathcal{F}_\theta(G), x_0, t, \psi(t, X))$ with initial condition $\psi(0, X) = X$. Therefore, we can generate the target sample by first randomly sampling $X' \sim \mathcal{N}(0, I)$, then solve the ODE with $X = X', t = 0$ until $t = 1$, and the target sample will be $\hat{\tau} = \psi(1, X')$. To solve ODE numerically, we discretize the ODE time domain into $N_s$ steps, and sample $t$ uniformly from $\{0, \frac{1}{N_s}, \frac{2}{N_s}, ..., 1\}$ in training. In testing, from random sample $X'$, we run Euler's method to solve ODE: $\psi(t_{i+1}, X') = \psi(t_i, X') + \frac{\mathcal{H}_\omega(\mathcal{F}_\theta(G), x_0, t_i, \psi(t_i, X'))}{N_s}$, where $t_i = \frac{i}{N_s}$, for $i = 0, 1..., N_s - 1$.

## 5. Experiments

**Implementation details.** We consider five robot simulation environments, including ZoneEnv in Jackermeier & Abate (2024) with **Linear** (single-integrator dynamics) and **Dubins** (Car dynamics), **PointMaze**, **AntMaze** (Fu et al., 2020) and **Franka Panda** robot arm (Gaz et al., 2019). The trajectory length is 512 for PointMaze and AntMaze and 64 for the other environments. For each robot domain, we start with the four STL templates discussed in Sec. 4.2, and for each template, we randomly generate 10000 STL specifications with 2 to 12 objects (regions) for goals and obstacles, and randomly initialize the agent location in each case. For simulation environments like Linear/Dubins and Franka Panda, we use a gradient-based solver to collect trajectories that maximize the STL robustness scores. For non-differentiable environments like PointMaze and AntMaze, we first plan for waypoints on the grids using A* search algorithm, then we use waypoint tracking controllers to generate the low-level

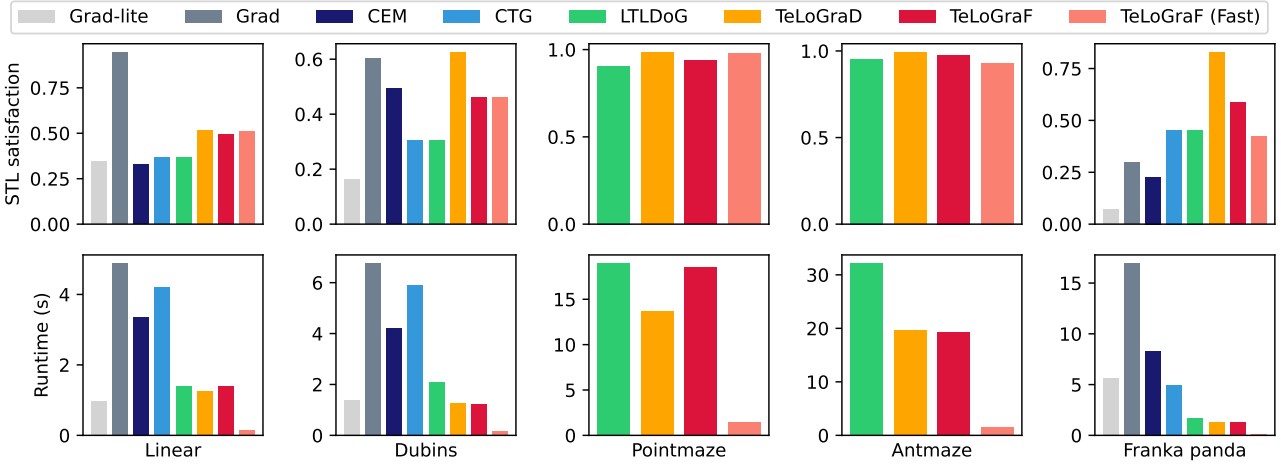

Figure 4: Main results. Our methods outperform both learning-based methods (**CTG** and **LTLDoG**) and classical methods (**Grad**, **CEM**). In four out of the five benchmarks, **TeLoGraD** achieves the highest solution quality. **TeLoGraF (Fast)** achieves the best trade-off between solution quality and efficiency over all benchmarks, especially being 123.6X faster than **Grad** and 60.7X faster than **CEM** with a higher satisfaction rate in the **Franka Panda** environment.

trajectories. For each STL, we generate 2 demonstrations, resulting in 80k trajectories under each robot domain. We use 80% for training and 20% for validation. Our TeLoGraF uses a GNN encoder with GCN layers (Kipf & Welling, 2016) with 4 hidden layers and 256 units in each layer, and a ReLU activation (Nair & Hinton, 2010) is used for the intermediate layers. The output embedding dimension is 256. The backbone network follows the UNet architecture used in (Janner et al., 2022) with two extra input for the STL embedding and the initial states. In the flow matching, we set the flow step $N_s = 100$. The learning pipeline is implemented in Pytorch Geometric (Fey & Lenssen, 2019; Paszke et al., 2019). The training is conducted for 1000 epochs with a batch size of 256. We use the commonly used ADAM (Kingma, 2014) optimizer with an initial learning rate $5 \times 10^{-4}$ and a cosine annealing schedule that reduces the learning rate to $5 \times 10^{-5}$ at the 900-th epochs and then keep it as constant for the rest 100 epochs. We use Nvidia L40S GPUs for the training, where each training job takes 6-24 hours on a single GPU. During evaluation, for each method, we sample 256 STL specifications for comparison (in Fig. 4 we sample on the validation set, whereas in Fig. 5 we sample on both training and validation sets).

**Baselines.** We compare with both classical and learning-based methods for STL specification planning. For the classical methods[3], we compare with **Grad**: a gradient-based method in Dawson & Fan (2022), **Grad-lite**: using the gradient-based method but with fewer iterations, and **CEM**: a sampling-based method (Kapoor et al., 2020). For the learning-based baselines, we compare with **CTG**: (Zhong

et al., 2023), which uses gradient guidance for the diffusion models, and **LTLDoG**: (Feng et al., 2024), which uses classifier guidance for the diffusion models (we change their LTL classifier to an STL classifier for our tasks). We also consider varied forms of encoder architectures, such as **GRU**: (Chung et al., 2014), which uses gated recurrent units to encode the sequence of STL formula, similar to Hashimoto et al. (2022), **Transformer**: (Vaswani, 2017), which uses attention-based auto-regressive models to encode the sequence of STL formula similar to GRU, and **TreeLSTM**: (Tai et al., 2015), which is similar to GNN but instead of synchronized message passing, TreeLSTM updates nodes features layer-by-layer via LSTM in a bottom-up fashion on syntax trees. Our method consists three variations, **TeLoGraD** (Diffusion): which uses GNN as encoder and learns the trajectories via diffusion models, **TeLoGraF**: which uses GNN as encoder and learns the trajectories via flow models, and **TeLoGraF (Fast)**: uses the pretrained TeLoGraF model with less ODE steps to generate samples.

**Metrics..** For each STL, we sample 1024 trajectories and pick the one with the highest STL score as the final trajectory. We compute the average STL satisfaction rate for the final trajectories (ratio of final trajectories that satisfy the STL) and the average computation runtime for the trajectory generation. Without special notice, we focused on the validation set STL satisfaction rate because the satisfaction rate in the training split is close to 1.

## 5.1. Main results

We first compare our method variations (TeLoGraD, TeLoGraF, and TeLoGraF (Fast)) with classical and

---

[3]We compare with classical methods only on selected differentiable environments.

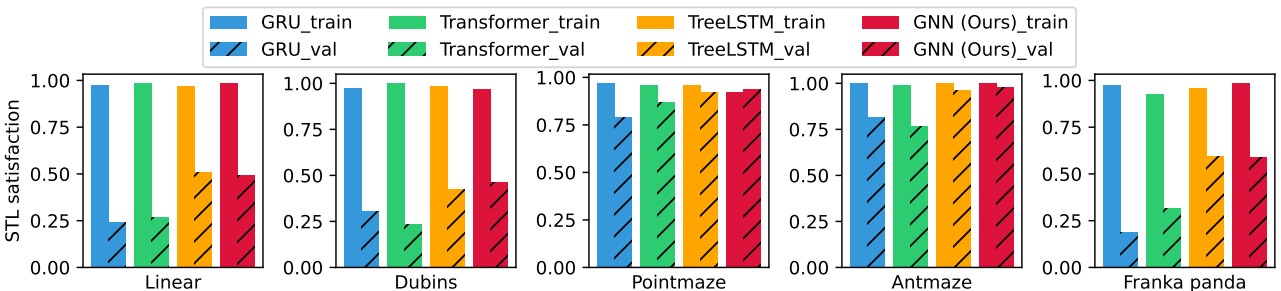

Figure 5: Different encoder architecture comparisons. GNN and TreeLSTM works the best on the validation set.

learning-based STL planning baselines. As shown in Fig. 4, on most of the environments, our methods achieve a better trade-off between solution quality and runtime. TeLoGraD achieves the highest performance regarding STL satisfaction, with the computation budget lower than almost all the baselines (except Grad-lite on Linear environment, but Grad-lite has low satisfaction rate), which first shows the efficacy of our designed pipeline. The advantage over classical methods increases as we move from low-dimension environments to the high-dimension Franka Panda environment, as both gradient-based and sampling-based methods struggles to provide high-quality solutions in a high-dimension space. Compared to learning-based methods such as CTG and LTLDoG, our methods do not compute the time-consuming guidance step in the inference stage, hence achieving better efficiency. Interestingly, the results under the maze environments (PointMaze and AntMaze) are very close among these learning-based methods, and the STL satisfaction rate is the highest among all environments. This is probably because the data distribution is limited to the maze layout, making learning easier. Among our methods, TeLoGraF and TeLoGraF(Fast) replace the diffusion models in TeLoGraD with flow-matching models, leading to a performance drop in Dubins and Franka Panda environments, which might be owed to the nonlinear dynamics or forward-kinematics under these environments. However, TeLoGraF (Fast) can achieve on par of TeLoGraF's satisfaction rate with only 1/10 of the runtime, making itself a super-efficient algorithm. On Franka Panda environment, TeLoGraF (Fast) is 123.6X faster than Grad and 60.7X faster than CEM with a higher satisfaction rate than both methods. Overall, the results demonstrate that TeLoGraD provides the best balance between solution quality and computational efficiency, while TeLoGraF (Fast) emerges as a promising alternative for real-time applications for the temporal logic task planning.

### 5.2. Ablation study on the flow steps

To understand how fast the flow model can be accelerated while maintaining the solution quality, we design a test on the Dubins environment with varied number of flow steps

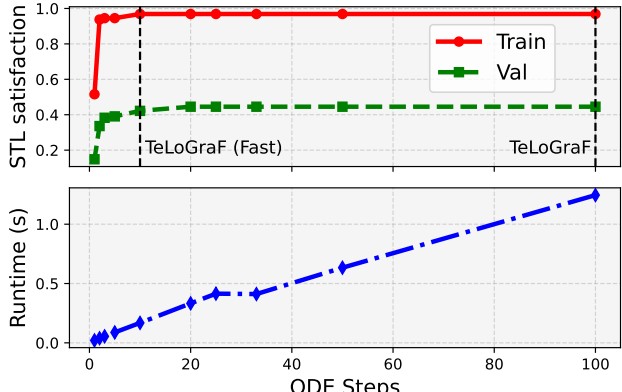

Figure 6: Ablation studies on ODE steps in Dubins environment. TeLoGraF(Fast) achieves a similar STL satisfaction compared to TeLoGraF with only 1/10 of its runtime.

for the sampling. The original TeLoGraF needs to run 100 ODE steps for data generation. Here, we reduce the ODE steps by increasing the ODE time step size[4] in each flow step. We evaluate the solution quality by checking its STL satisfaction. As shown in Fig. 6, the algorithm runtime grows linearly with the number of ODE steps. The STL satisfaction rate for the original TeLoGraF is 0.97 on the training split and 0.45 on the validation split. The scores do not drop until the number of ODE steps decreases to 10. Even with 2 ODE steps, the model can obtain an STL satisfaction rate of 0.94 on the training split and 0.34 on the validation. These observations demonstrate the sampling efficiency of the flow models and we hence use 10 ODE steps for "TeLoGraF (fast)".

### 5.3. Ablation study on encoder designs for STL

We compare different architectures to encode STL syntax, including sequence models like GRU and Transformer, and

---

[4]This ODE time step size should be distinguished with the algorithm runtime. The ODE time interval is fixed from 0 to 1 in the flow model sampling process. Thus, the step size in each ODE step determines the number of ODE steps needed.

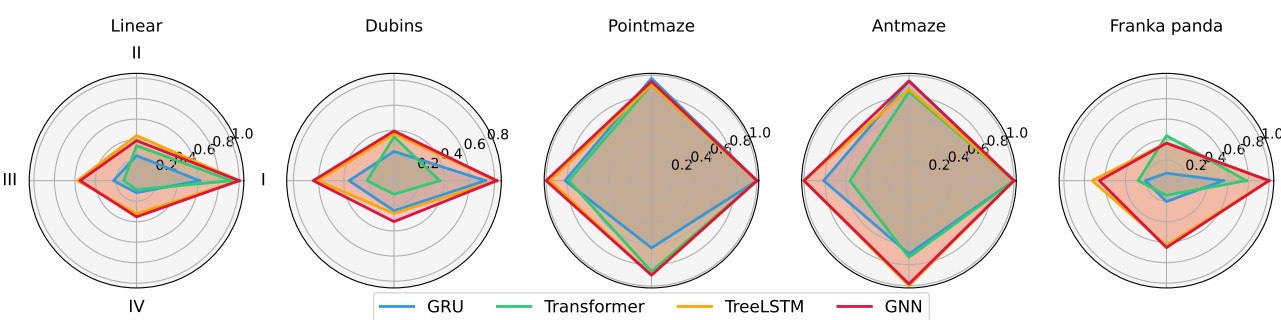

Figure 7: STL satisfaction rate distribution for different categories (I. single-goal; II. multi-goal; III. sequential; IV. partial).

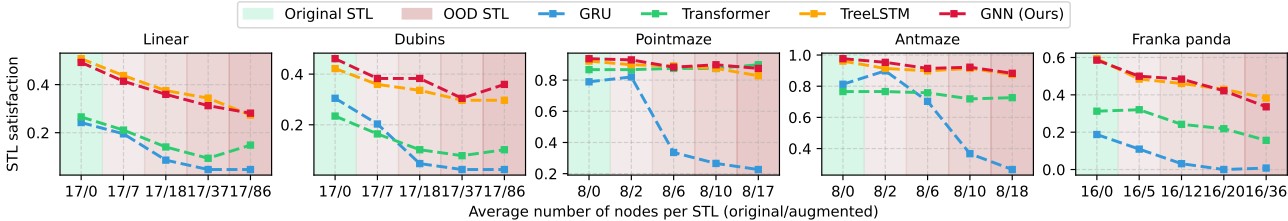

Figure 8: Encoder robustness test under varied STL augmentation (shaded in red). Graph-based encoders (GNN, TreeLSTM) demonstrate greater resilience to synthetic modifications compared to sequence-based models (GRU, Transformer).

graph-encoding: GNN and TreeLSTM. We use the same pipeline as TeLoGraF, and only change the encoder network. As shown in Fig. 5, all the encoders have a close to $100\%$ satisfaction rate on the training splits, indicating they can distinguish different STL syntax. However, their abilities to generalize to unseen STL syntax are different: GRU and Transformer get lower STL scores than TreeLSTM and GNN; one reason could be that the graph-encoding models inherit permutation-invariance of the syntax tree ("A$\vee$B" is the same as "B$\vee$A"), making the learning more efficient. The performance of TreeLSTM and GNN are similar, and we select GNN encoding owing to its brevity.

### 5.4. Encoder coverage analysis on varied STL types

We show the validation STL satisfaction over different STL categories in Fig. 7. As expected, most encoders can lead to high satisfaction rate on single-goal STLs. The improvements of the graph-encoding over sequence models are mainly from "Sequential" and "Partial" types, which implies that, beyond permutation-invariance, graph-encoding methods also excel at propagating time constraints and temporal orderings than the sequence models. However, all models' performance on II-IV STLs is relatively low on Linear, Dubins and Franka Panda, which implies it is challenging for current models to generalize to complex STLs such as multi-goal satisfaction, sequential, or partial constraints. This highlights the value of our dataset in examining the limitation of the current models in handling complex STLs. Future improvements in architectures or training strategies

are necessary to foster better temporal logic planning.

### 5.5. Robustness test for encoders on out-domain STLs

We alter the STLs in the validation splits and use the pretrained models on the original dataset to produce trajectories. We deliberately keep the trajectories fixed and only change the STLs to ensure the backbone is not affected by out-of-distribution trajectories. For each $\wedge$ and $\vee$ operator in the STL syntax tree, we randomly duplicate a number of their children nodes (e.g., "A$\wedge$B" $\rightarrow$ "A$\wedge$B$\wedge$B"). This augmentation will not affect the solutions. Ideally, if the model is robust, the output should have the same STL satisfaction rate. As shown in Fig. 8, all models result in degradation as the number of nodes augmented increases, but the graph-based encoder (GNN, TreeLSTM) is resilient to the augmentation. In most out-domain cases, they can even outperform the sequence models with normal STL inputs. This implies graph-based encoders inherently capture the structural STL more effectively and generalize better under augmentations.

**Limitations.** First, TeLoGraF is data-driven and does not offer soundness and completeness guarantees as other classical planning methods. Its performance degrades for tasks with complex STL syntax or STLs that are heavily out-ofdistribution. This limitation is inherent to most learningbased methods, as they rely on the distribution of training data. Trajectory refinement with classical methods could enhance the model's generalization and robustness on these temporal logic tasks. Besides, though TeLoGraF can predict "state-action" trajectories, in inference we only use

its predicted waypoints rather than rolling out trajectories based on the predicted action sequence. Ensuring dynamic consistency can be achieved via tracking controllers (Sun et al., 2022) or inverse dynamics (Ajay et al., 2022), we leave these for future work.

# 6. Conclusion

We propose TeLoGraF, a novel learning-based framework for solving general Signal Temporal Logic (STL) tasks via graph-encoding and flow-matching. TeLoGraF can handle flexible forms of specifications and outperforms existing classical and data-driven baselines while maintaining fast inference speed. Our analysis highlights the value of graph-based STL encoding. Aside from TeLoGraF, we introduce a new dataset for temporal logic planning, addressing the lack of diverse and structured STL benchmarks. Future research directions include improving the model's generalization and robustness towards more complex temporal logic structure and out-of-distribution STL specifications.

# Acknowledgements

This work was supported in part by the National Science Foundation (NSF) under Grant #CCF-2238030 and in part by ONR under Award N00014-22-1-2478. Any opinions, findings, conclusions, or recommendations expressed in this publication are those of the authors and don't reflect the views of the sponsors.

# Impact Statement

This paper presents work whose goal is to advance the field of robotics and decision-making for long-horizon tasks. There are some potential societal consequences of our work, for example, a fully-autonomous robot using our designed algorithm for decision-making could lead to potentially unsafe behavior. Thus, a safety-filter or backup policy for the planned behavior is needed for deployment. No other aspects which we feel must be specifically highlighted here.

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

# A. Experiment setups

## A.1. Simulation Environments

### A.1.1. LINEAR

We use a single-integrator dynamic model. The 2D state $(x, y)$ represents the 2D coordinates on a $xy$-plane, and the 2D control input $(u, v)$ reflects the velocities in these two directions. We set the time step duration $\Delta t = 0.5$. The system dynamic is described as:

$$\begin{cases} x_{t+1} & = x_t + u_t \Delta t \\ y_{t+1} & = y_t + v_t \Delta t \end{cases} \tag{6}$$

In this environment, we randomly generate goal regions and obstacles in a circle shape, and generate STL specifications using procedural methods. Finally, we use a gradient-based trajectory optimization method implemented in PyTorch (Paszke et al., 2019) to compute the demonstration trajectories. The loss function is constructed as:

$$\mathcal{L} = \min(0.5 - \rho(\tau, 0, \phi), 0) + c_1 \frac{1}{T} \sum_{t=0}^{T} \{\min(u_t^2 - 1, 0) + \min(v_t^2 - 1, 0)\} + c_2 \frac{1}{T} \sum_{t=0}^{T} (u_t^2 + v_t^2) \tag{7}$$

The first loss term maximizes the truncated robustness score $\rho$ for the trajectory $\tau = (x_0, u_0, ..., u_{T-1}, x_T)$ to ensure STL rule satisfaction for the STL $\phi$. The second loss term regulates the magnitude of the control input to be within 1. The third loss term minimizes the control magnitude to make the trajectory smooth.

### A.1.2. DUBINS

We use a car-like dynamic model. The 4D state $(x, y, \theta, v)$ represents the 2D coordinates on the $xy$-plane, the heading angle of the robot and the velocity of the robot, respectively. The 2D control input $(\omega, a)$ represents the angular velocity and the acceleration. We set the time step duration $\Delta t = 0.5$. The system dynamic is described as:

$$\begin{cases} x_{t+1} & = x_t + v_t \cos(\theta_t) \Delta t \\ y_{t+1} & = y_t + v_t \sin(\theta_t) \Delta t \\ \theta_{t+1} & = \theta_t + \omega_t \Delta t \\ v_{t+1} & = v_t + a_t \Delta t \end{cases} \tag{8}$$

In this environment, we randomly generate goal regions and obstacles in a circle shape, and generate STL specifications using procedural methods. Finally, we use a gradient-based trajectory optimization method implemented in PyTorch (Paszke et al., 2019) to compute the demonstration trajectories. The loss is similar to Eq. (7).

### A.1.3. POINTMAZE

We adapt PointMaze and AntMaze from these two environments from (Fu et al., 2020). The original environments consist of 2D navigation tasks where an agent must navigate a maze to reach a goal while avoiding obstacles. The dynamic model is internally handled by MuJoCo (Todorov et al., 2012), and we use the Gymnasium-Robotics package (de Lazcano et al., 2024) for simulation to collect data. We use the largest maze in both simulations with horizons set as 512 steps. In Pointmaze, the agent is modeled as a simple 2D point mass with position $(x, y)$ and the control input is the linear force along xy directions. We collect the 4-dimension trajectory with 2d xy-coordinate and 2d control input. To collect trajectories, we first generate the skeleton of the STL syntax but leave the time intervals as placeholders; then, we use a planning algorithm to find waypoints and then use a PD goal-reaching controller to collect trajectories. After the generation of the trajectories, we first ensure it reaches the final destination, and then we evaluate the robustness score on the trajectory without considering the time intervals. If the robustness score is positive, we randomly infer the possible time intervals for the goal-reaching sub-formulas and run post-verification to ensure the trajectory satisfies this STL.

### A.1.4. ANTMAZE

In AntMaze, the agent is modeled as a more complex 8-DoF quadruped robot with 27-dimension observation (1-dimension for z-coordinate of the torso, 4-dimension for the torso orientation (in Quaternion representation), 3-dimension velocity and 3-dimension angular velocity for the torso, 8-dimension for each of the joints' angles and another 8-dimension for each of

the joints' angular velocities), 8-dimension control input serving as the torques applied to each of the 8 joint rotors, and an extra 2-dimension for the xy-coordinates of the torso. We collect the 10-dimension trajectory with 2d xy-coordinate and 8-dimension control input. We adopt similar data collection methods as in PointMaze, with the difference of switching from PD controller to a RL goal-reaching policy.

### A.1.5. FRANKA PANDA

We use a 7 DoF Franka Panda robot arm model and conduct the simulation via PyBullet Simulator (Coumans & Bai, 2021). We use the standard URDF files to construct the scene: a table is generated via "table.urdf" at location $[0.4, 0, 0]$ with Euler angle $[0, 0, np.pi/2]$. A Franka Panda robot is generated via "franka/panda.urdf" at location $[0, 0, 0.6]$ with Euler angle $[0, 0, 0]$. The 7DoF joint angles for the robot are initialized at (in radian) $[0.0000, 0.0000, 0.0000, -2.0000, 0.0000, 1.8675, 0.0000]$ and are clipped from $[-2.8972, -1.7628, -2.8972, -3.0718, -2.8972, -0.0175, -2.8972]$ to $[2.8972, 1.7628, 2.8972, -0.0698, 2.8972, 3.7525, 2.8972]$ by following the guidelines. Random goal objects are generated on the table whereas the obstacles are generated both on the table and in the midair to increase the task difficulty. We use a gradient-based method to plan the trajectory to satisfy the STL conditions. We use pytorch-kinematics (Zhong et al., 2024) library to leverage Pytorch and GPU devices to compute the forward kinematic in a parallelized and efficient way. The time horizon is 64 with the time step duration $\Delta t = 0.05$.

### A.2. Procedural methods to generate STL formulas

We adopt procedural methods to generate the STL formulas. We first assign the type of STL we want to generate (Single-goal, multi-goal, sequential, partial-order). Then each formula is augmented with randomly chosen 0-6 obstacles and 0-4 goals while they are not colliding with each other. Next, we spawn the agent position in the free space and use either search-planning-then-tracking method or end-to-end gradient-based method to collect the trajectories. More details are in the code, and here in the following section we will show a subset of demonstrations with the visualized STL syntax trees.

### A.3. Baselines

#### A.3.1. GRAD/GRAD-LITE

We directly use the data collection method we have for the three benchmarks, Linear, Dubins, and Panda. We set different gradient-descent iterations for the different environments. For Linear and Dubins, we set niters=10 for "Grad-lite" and niters=50 for "Grad", and for Franka Panda environment, we set niters=50 for "Grad-lite" and niters=150 for "Grad".

#### A.3.2. CEM

We use the Cross Entropy Method with the elite size of 25, the population sample size of 64, and the number of iteration steps is set to 100.

#### A.3.3. CTG

We followed the guidance in Zhong et al. (2023) and implemented this baseline by ourselves, as the original baseline is conducted for the autonomous driving dataset. To ensure we don't have STL-condition-input encountered in training and to make a fair comparison with TeLoGraF backbone, we train a goal-conditioned flow-matching policy where the goal condition is constructed as a set of subgoals and obstacles (we also mark whether they are goals or obstacles in this embedding). At the inference stage, we use the gradient-based method to construct the guidance for the flow-matching model.

#### A.3.4. LTLDOG

Similar to CTG, we first train a goal-conditioned flow-matching policy. Further, we train an STL classifier as suggested in Feng et al. (2024). At each iteration, in the minibatch data we add noise to the demonstration data and compute their robustness score (they are most likely to be negative scores) for the corresponding STL. We also save the STL score for the existing demonstration trajectories. We train our STL classifier to predict the scores for both the original trajectories and the perturbed trajectories using a simple L2 loss (suggested in Feng et al. (2024)). With 50 epochs of training, on most benchmarks the STL classifier can already achieve more than 90% accuracy in classifying positive and negative sample trajectories regarding the STL specification (and since we need to evaluate the robustness score during training, the training

is 10-20X slower than the flow-matching model training). We then use this trained STL classifier in the flow-matching generation process as guidance.

## B. Additional visualizations from our flow-matching model

Here we show the TeLoGraF generated trajectories on the Dubins benchmark and show how the trajectory distribution and satisfaction results change as the number of ODE steps decrease from 100 (the original setup for TeLoGraF) to 1. We plot the goals in blue color, obstacles in gray color, the initial state is plotted in a marker shape. The green trajectories are the demonstration data, and the red ones are the flow-model generated trajectories (we only plot 4 of them on each plot), where the dark red one is the generated trajectory with the highest robustness score for this STL. We can see that as the number of ODE steps decreases, the trajectories set become more "concentrated". The predicted trajectory can always satisfy the STL, until the last one (ODE steps=1) where the trajectory fails to reach goal-1, leading to a failure case.

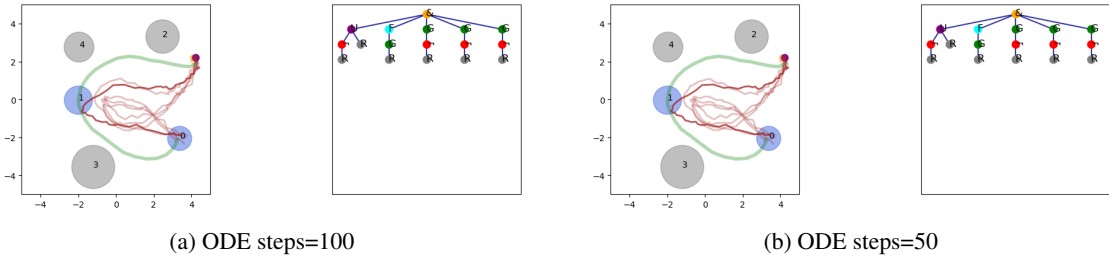

(a) ODE steps=100        (b) ODE steps=50

Figure 9: Varied ODE steps, Dubins environment, TeLoGraF

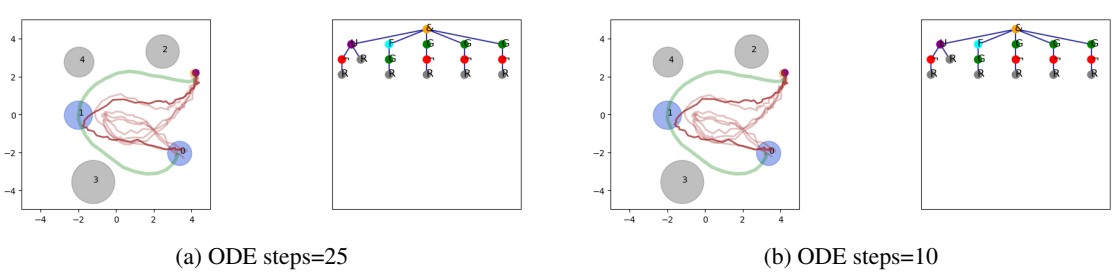

(a) ODE steps=25        (b) ODE steps=10

Figure 10: Varied ODE steps, Dubins environment, TeLoGraF

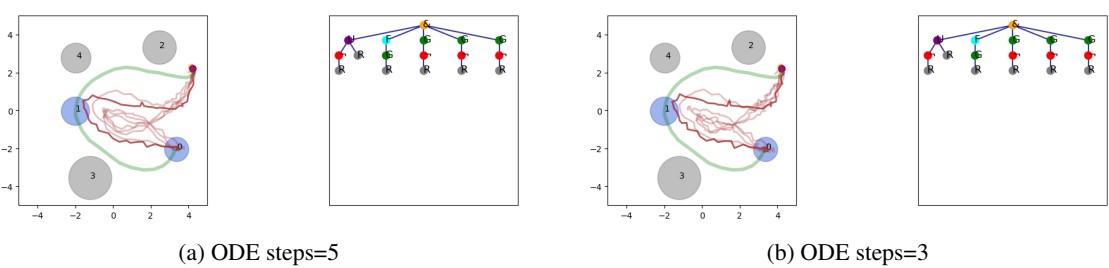

(a) ODE steps=5        (b) ODE steps=3

Figure 11: Varied ODE steps, Dubins environment, TeLoGraF

## C. Additional visualizations from demonstration dataset

The results shown here are in the raw version (without data cleaning or improved visibility). Based on these data, we randomly select 2 demonstrations per STL listed in the dataset and collect 40K STLs for each of the benchmarks, gathering in total 200K STLs with 400K trajectories.

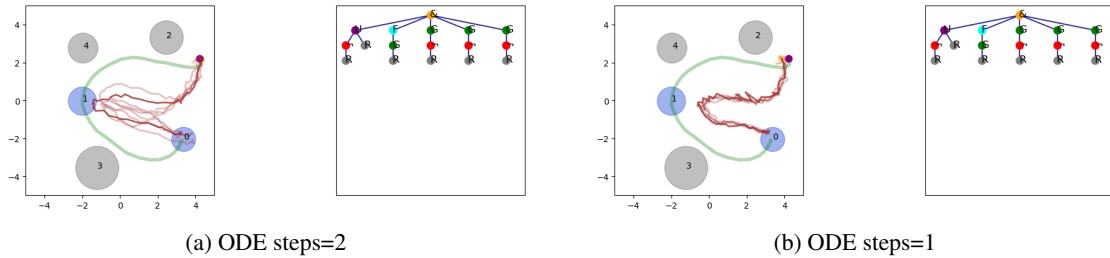

(a) ODE steps=2  (b) ODE steps=1

Figure 12: Varied ODE steps, Dubins environment, TeLoGraF

## C.1. Linear environment

Here in the Linear environment, for one given STL, we actually start from eight different initial locations and initialize with 8 random control sequences for each initial location. The results satisfying the STL conditions (green) will be saved and the red ones are filtered out.

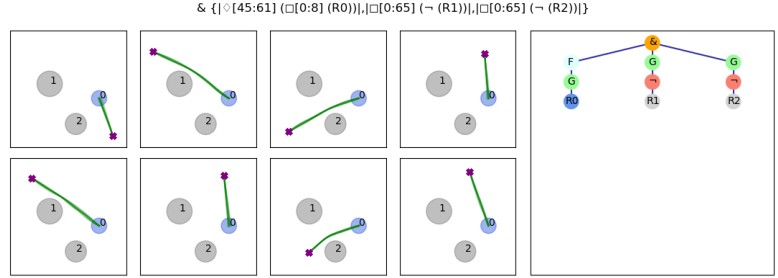

Figure 13: Linear environment. Single-goal STL

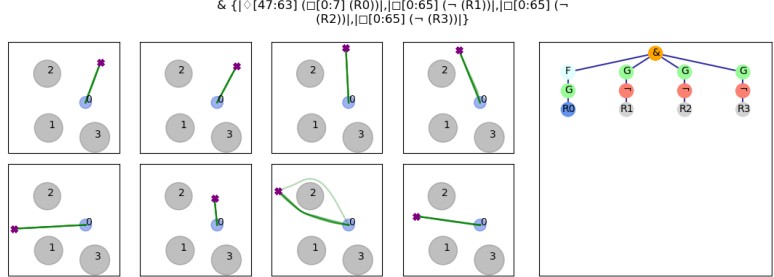

Figure 14: Linear environment. Single-goal STL

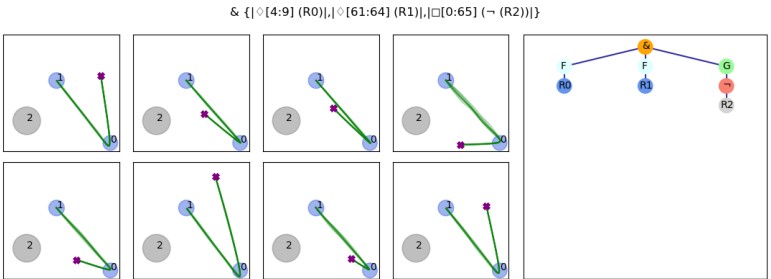

Figure 15: Linear environment. Multi-goal STL

& {|(◇[55:57] (R0)) | (◇[59:62] (□[0:10] (R1)))|,|□[0:65] (¬ (R2))|,|□[0:65] (¬ (R3))|,|□[0:65] (¬ (R4))|,|□[0:65] (¬ (R5))|,|□[0:65] (¬ (R6))|}

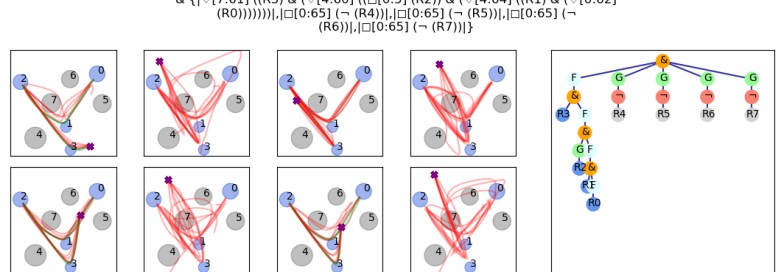

Figure 16: Linear environment. Multi-goal STL

& {|◇[7:61] ((R3) & (◇[4:60] ((□[0:5] (R2)) & (◇[4:64] ((R1) & (◇[0:62] (R0)))))))|,|□[0:65] (¬ (R4))|,|□[0:65] (¬ (R5))|,|□[0:65] (¬ (R6))|,|□[0:65] (¬ (R7))|}

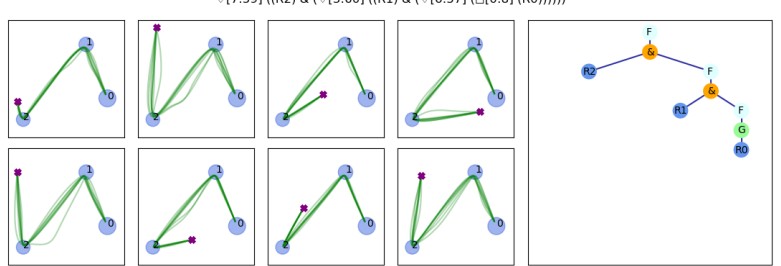

Figure 17: Linear environment. Sequential STL

◇[7:59] ((R2) & (◇[3:60] ((R1) & (◇[6:57] (□[0:8] (R0))))))

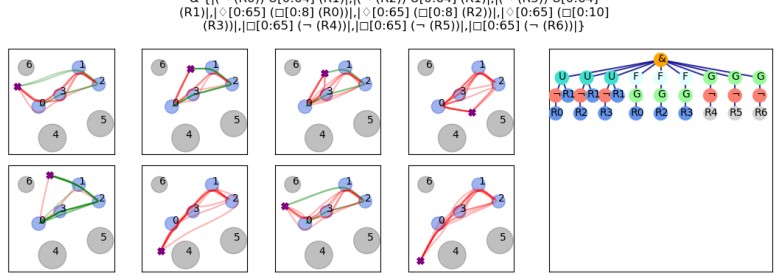

Figure 18: Linear environment. Sequential STL

& {|(¬ (R0)) U[0:64] (R1)|,|(¬ (R2)) U[0:64] (R1)|,|(¬ (R3)) U[0:64] (R1)|,|◇[0:65] (□[0:8] (R0))|,|◇[0:65] (□[0:8] (R2))|,|◇[0:65] (□[0:10] (R3))|,|□[0:65] (¬ (R4))|,|□[0:65] (¬ (R5))|,|□[0:65] (¬ (R6))|}

Figure 19: Linear environment. Partial STL

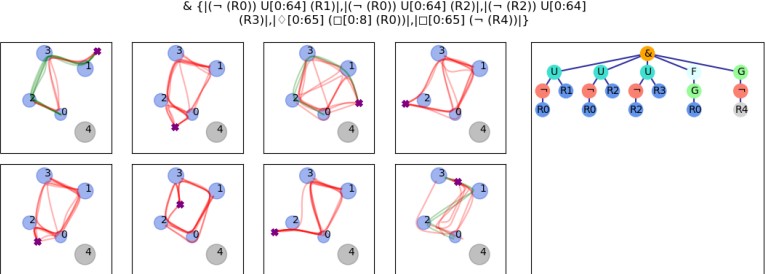

Figure 20: Linear environment. Partial STL

## C.2. Dubins environment

In Dubin's environment, the collected demonstration trajectories are more diverse and drastically change over time, owing to the large angular velocity and the acceleration rate.

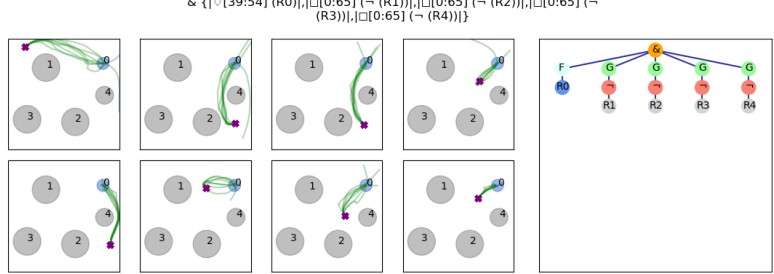

Figure 21: Dubins environment. Single-goal STL

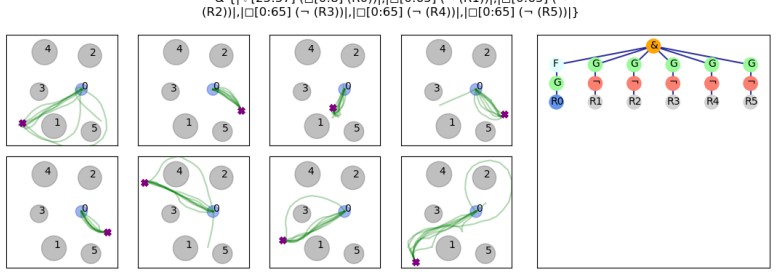

Figure 22: Dubins environment. Single-goal STL

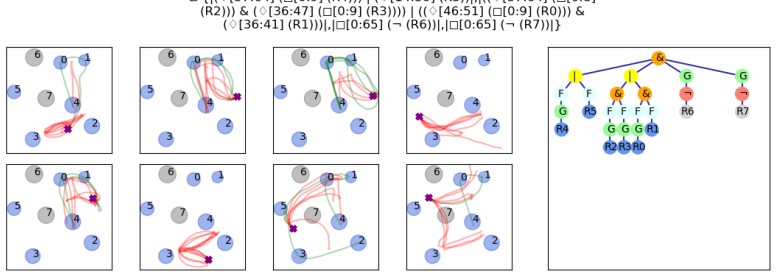

Figure 23: Dubins environment. Multi-goal STL

& {|◇[30:49] (□[0:5] (R0))|,|◇[13:41] (R1)|,|□[0:65] (¬ (R2))|,|□[0:65] (¬ (R3))|}

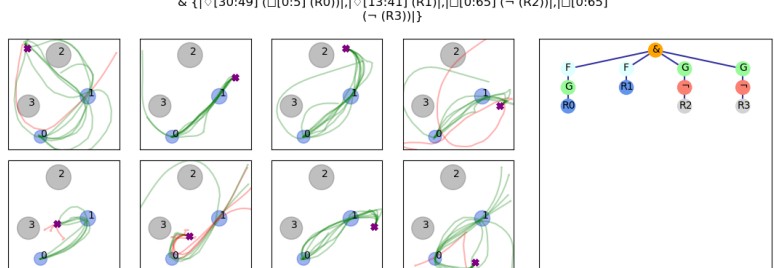

Figure 24: Dubins environment. Multi-goal STL

◇[7:60] ((R3) & (◇[4:63] ((R2) & (◇[7:59] ((□[0:10] (R1)) & (◇[1:60] (R0))))))))

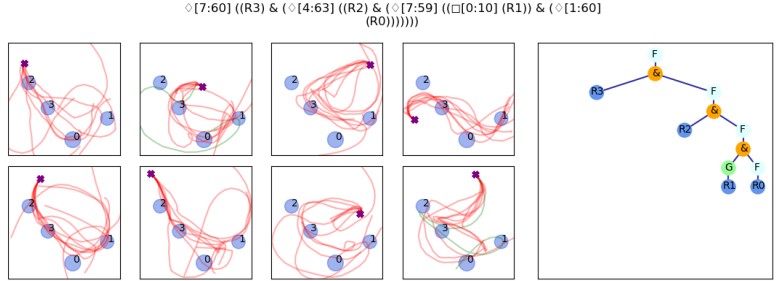

Figure 25: Dubins environment. Sequential STL

& {|◇[2:64] ((□[0:6] (R3)) & (◇[1:57] ((□[0:9] (R2)) & (◇[7:63] ((□[0:4] (R1)) & (◇[7:57] (R0)))))))|,|□[0:65] (¬ (R4))|,|□[0:65] (¬ (R5))|,|□[0:65] (¬ (R6))|}

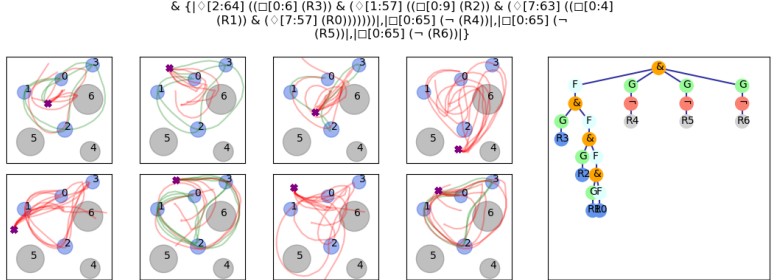

Figure 26: Dubins environment. Sequential STL

& {|(¬ (R0)) U[0:64] (R1)|,|(¬ (R2)) U[0:64] (R3)|,|(¬ (R4)) U[0:64] (R3)|,|◇[0:65] (□[0:4] (R0))|,|◇[0:65] (R2)|,|◇[0:65] (R4)|,|□[0:65] (¬ (R5))|}

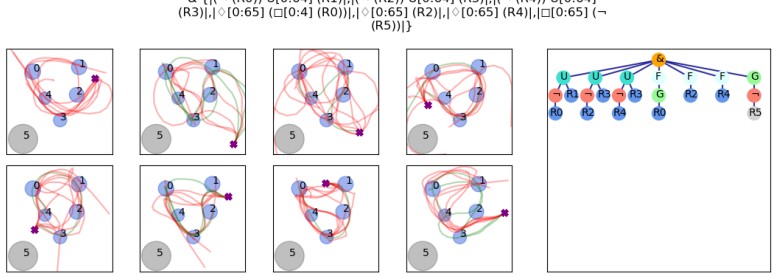

Figure 27: Dubins environment. Partial STL

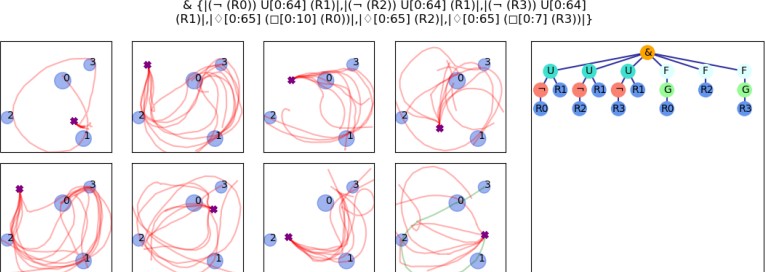

Figure 28: Dubins environment. Partial STL

## C.3. PointMaze

Here, the pink trajectories are the demonstration data collected.

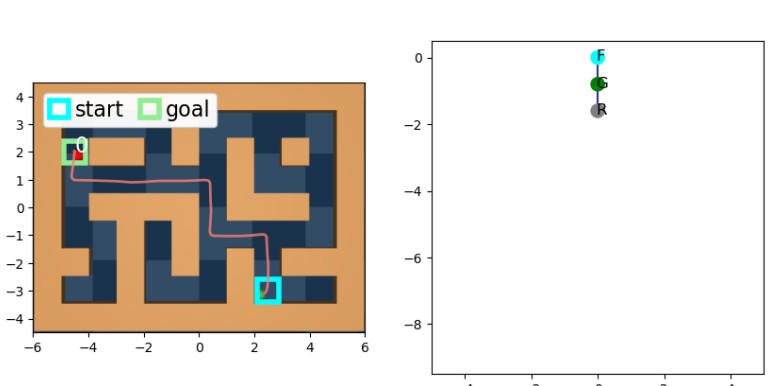

Figure 29: PointMaze environment. Single-goal STL

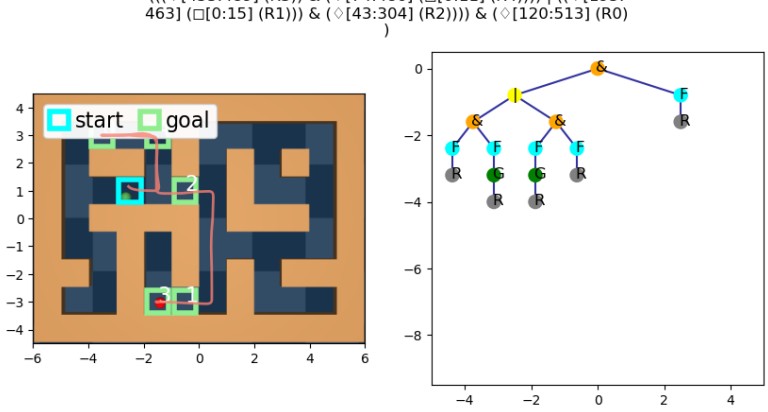

Figure 30: PointMaze environment. Multi-goal STL

## C.4. AntMaze

Here, the pink trajectories are the demonstration data collected.

(◇[203:496] (□[0:4] (R0))) | (◇[73:444] (R1))

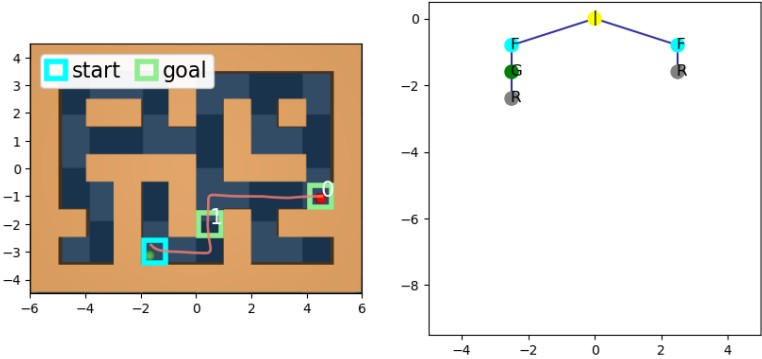

Figure 31: PointMaze environment. Multi-goal STL

◇[0:178] ((□[0:11] (R1)) & (◇[0:365] (R0)))

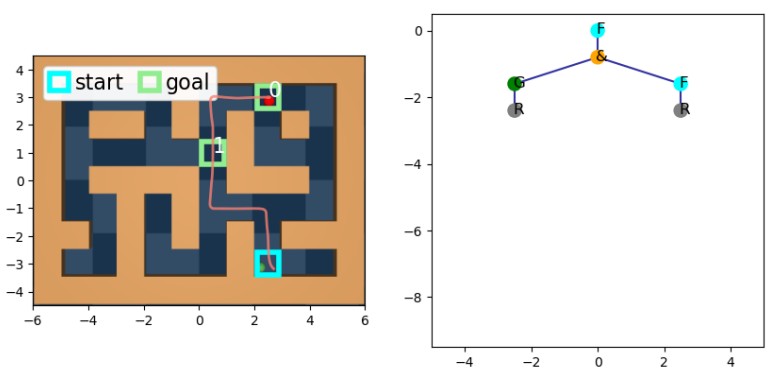

Figure 32: PointMaze environment. Sequential STL

◇[0:195] ((□[0:11] (R2)) & (◇[0:87] ((R1) & (◇[0:407] (R0)))
))

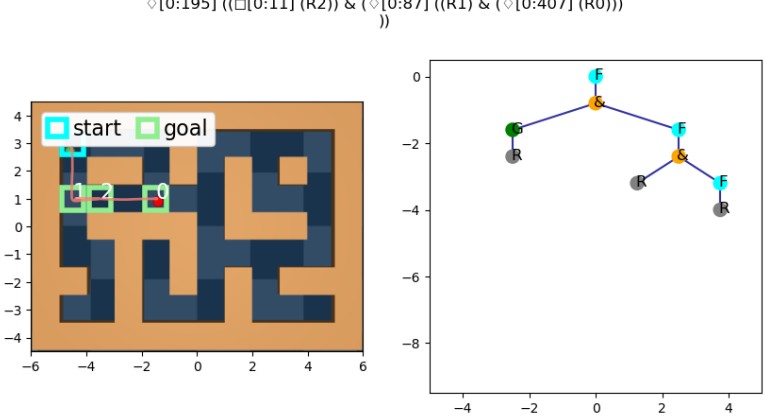

Figure 33: PointMaze environment. Sequential STL

◇[0:193] ((R1) & (◇[0:370] (R0)))

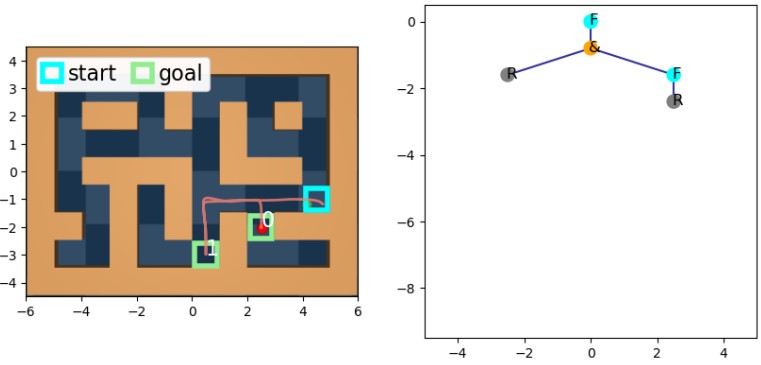

Figure 34: PointMaze environment. Sequential STL

◇[0:51] ((R3) & (◇[0:167] ((R2) & (◇[0:70] ((R1) & (◇[0:320] (R0)))))))

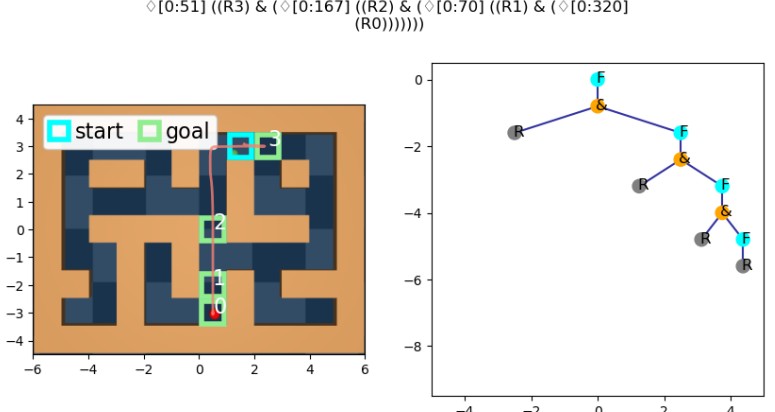

Figure 35: PointMaze environment. Sequential STL

& {|(¬ (R4)) U1[0:513] (R0)|,|(¬ (R5)) U1[0:513] (R1)|,|(¬ (R6)) U1[0:513] (R2)|,|(¬ (R7)) U1[0:513] (R3)|,|◇[341:351] (□[0:7] (R8))|}

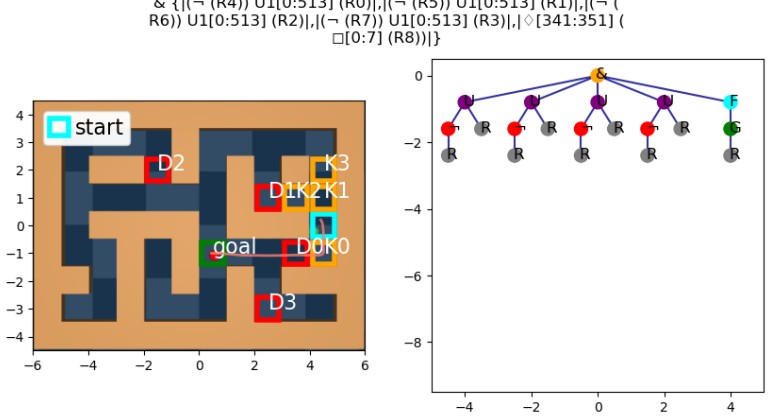

Figure 36: PointMaze environment. Partial STL

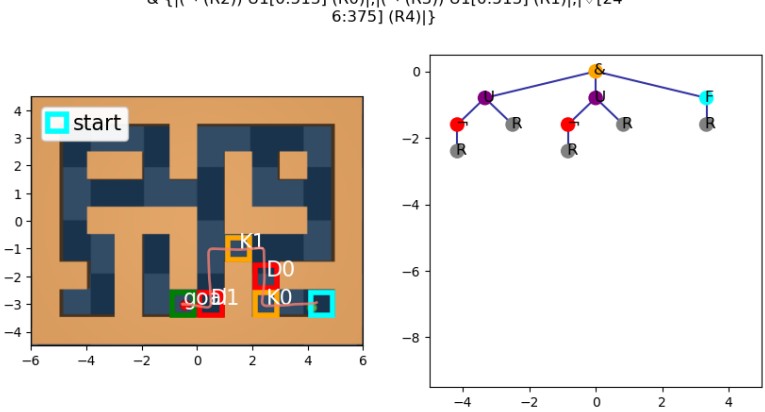

Figure 37: PointMaze environment. Partial STL

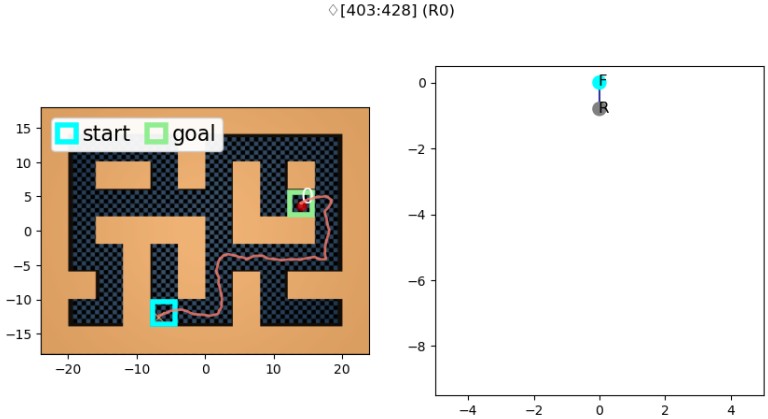

Figure 38: AntMaze environment. Single-goal STL

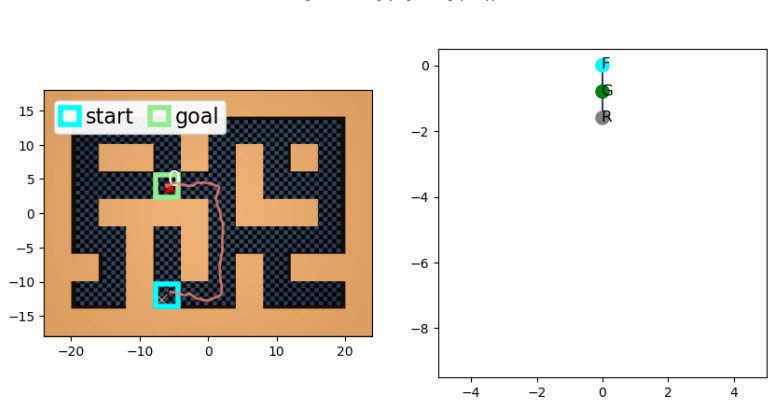

Figure 39: AntMaze environment. Single-goal STL

| {|◇[332:431] (R0)|,|◇[207:418] (R1)|,|◇[439:513] (□[0:10] (R2))|}

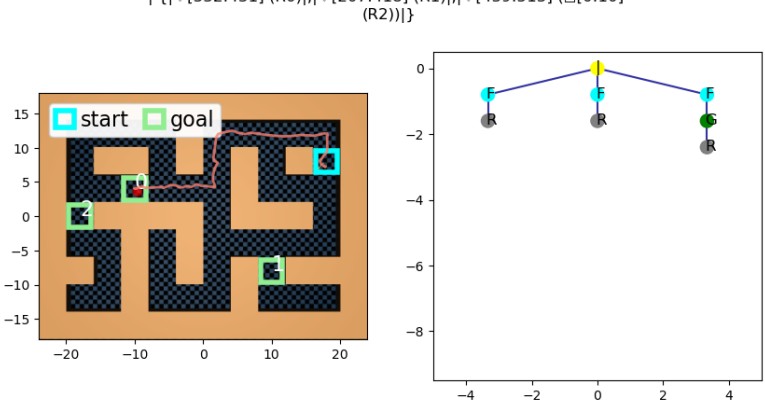

Figure 40: AntMaze environment. Multi-goal STL

(((◇[0:513] (□[0:39] (R5))) & (◇[0:513] (□[0:23] (R6)))) | ( ◇[334:384] (R4))) & (((◇[99:136] (R2)) & (◇[0:513] (R3))) | ((◇[208:247] (R0)) & (◇[141:178] (□[0:8] (R1)))))

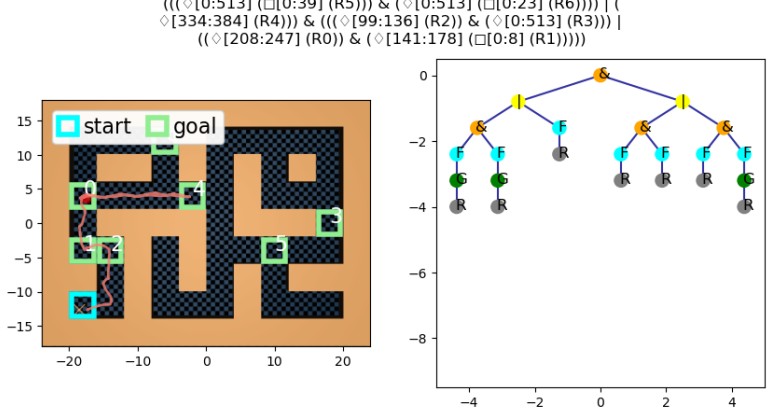

Figure 41: AntMaze environment. Multi-goal STL

◇[0:117] ((R1) & (◇[0:456] (□[0:216] (R0))))

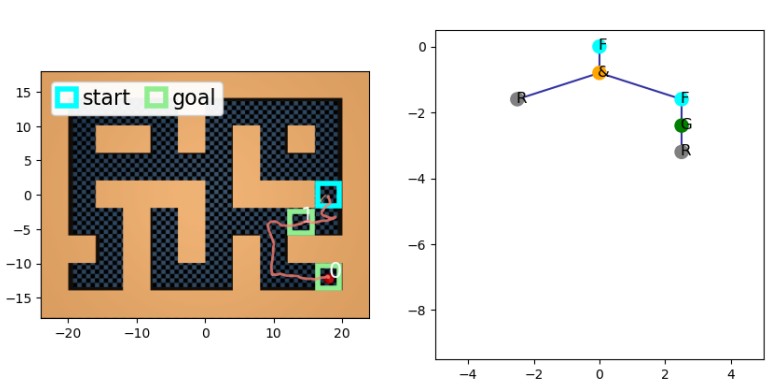

Figure 42: AntMaze environment. Sequential STL

◇[0:140] ((R2) & (◇[0:139] ((R1) & (◇[0:348] (R0)))))

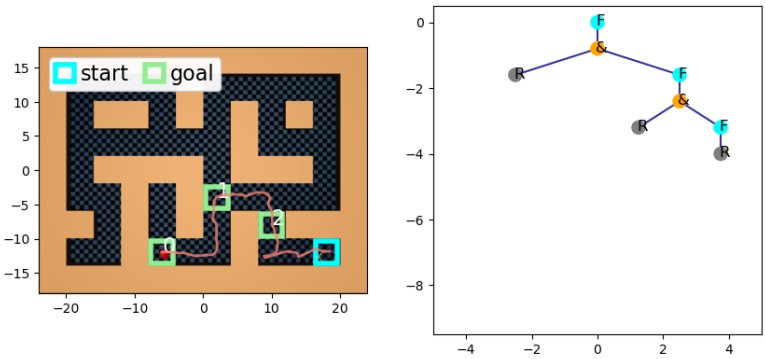

Figure 43: AntMaze environment. Sequential STL

& {|(¬ (R2)) U1[0:513] (R0)|,|(¬ (R3)) U1[0:513] (R1)|,|◇[50 8:511] (□[0:2] (R4))|}

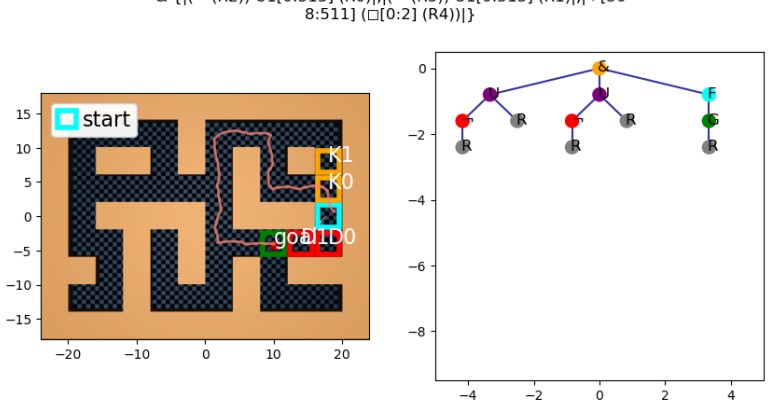

Figure 44: AntMaze environment. Partial STL

& {|(¬ (R3)) U1[0:513] (R0)|,|(¬ (R4)) U1[0:513] (R1)|,|(¬ ( R5)) U1[0:513] (R2)|,|◇[438:473] (R6)|}

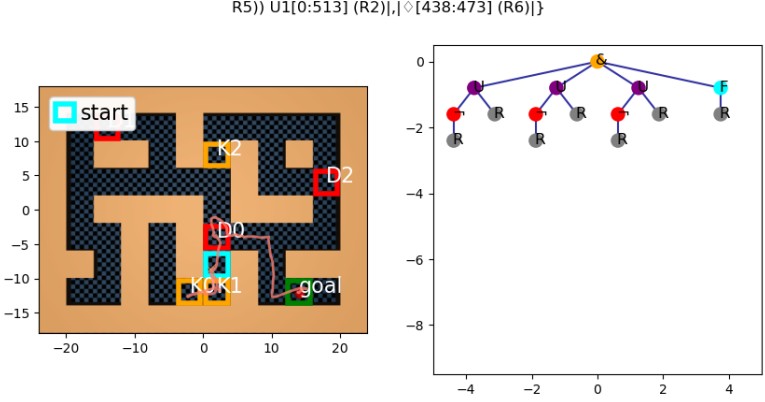

Figure 45: AntMaze environment. Partial STL

## C.5. Franka panda environment

Here, the blue trajectories in the last subfigure are the demonstrations that were collected.

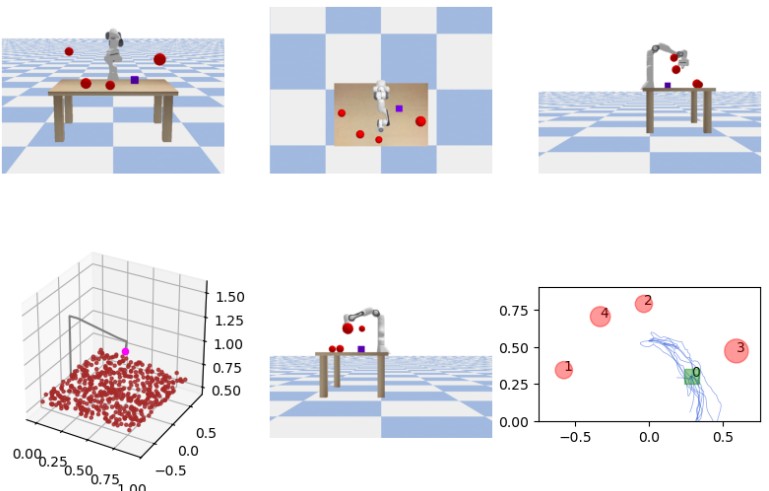

Figure 46: Franka Panda environment. Single-goal STL

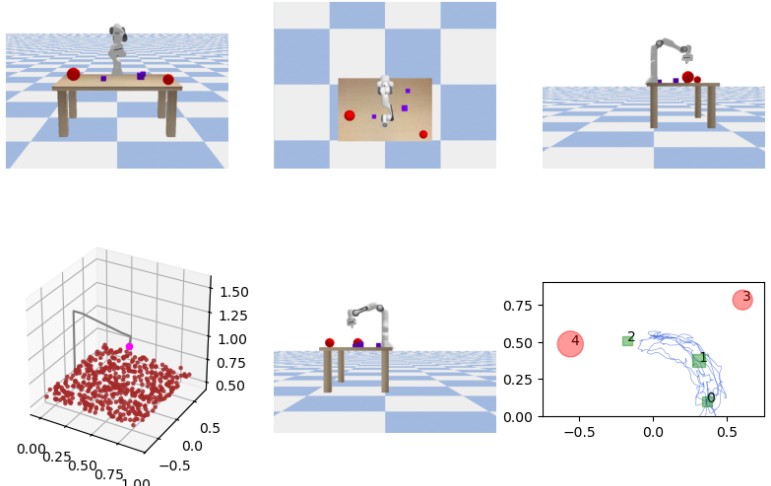

Figure 47: Franka Panda environment. Multi-goal STL

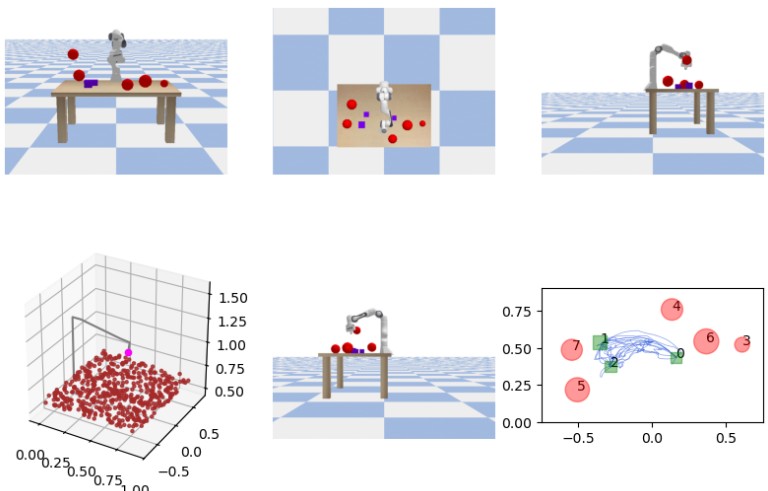

Figure 48: Franka Panda environment. Sequential STL

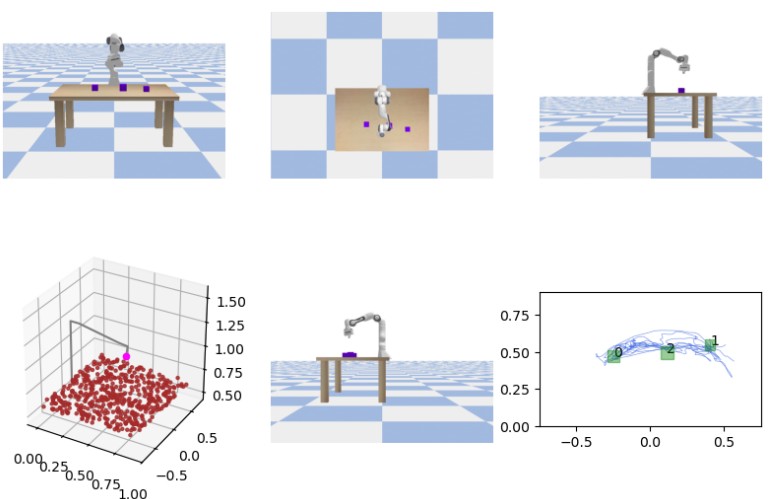

Figure 49: Franka Panda environment. Partial STL

