# OpenReview forum: "TeLoGraF: Temporal Logic Planning via Graph-encoded Flow Matching"
_ICML.cc/2025/Conference — ICML 2025 poster_

### Official Review · Reviewer_8aYy · 2025-03-03

**Overall Recommendation:** 3

**Summary:**

The paper studies the problem of training an RL/planning agent that takes as input (i) a Signal Temporal Logic (STL) specification and (ii) a start state and then generates a plan/trajectory in the environment that starts at the given start state and satisfies the given STL specification. The proposed learning algorithm involves (i) sampling STL specifications from four main categories that appear commonly in practice, (ii) sampling expert trajectories that satisfy each specification (computed using well-known but slow optimization procedures), and (iii) using the sampled dataset of specification-trajectory pairs to fit a generative model that generates trajectories given a specification and a start state. The generative model architecture involves two main components: (i) a graph neural network architecture to encode the given STL specification and (ii) a temporal U-Net flow model that can be used to generate a trajectory given the encoding of the specification. Experiments in complex environments indicate that the proposed approach can outperform baselines w.r.t. satisfaction rate under compute constraints.

**Claims And Evidence:**

The high-level claims are supported by decent evidence in the paper. For instance, the authors show that the chosen encoder architecture outperforms alternatives via a sufficiently thorough ablation study. The only claim that seems slightly problematic is that authors are the first to train a generative model for planning conditioned on STL specifications. Although it appears that this claim is true, I am not sure if this is a significant claim given there is already work on training RL agents that output actions conditioned on LTL specifications (see below).

**Essential References Not Discussed:**

I think the below reference [1] is very similar to this paper. The approach presented in [1] also involves using a GNN to encode the temporal specification (LTL). Although this paper differs slightly since the focus is on using a generative model for planning, it is still very similar to [1] which trains an actor network instead of generative model. The main differences seem to be (i) using STL instead of LTL and (ii) using a generative model for planning instead of training an actor network. These differences are fairly minor and [1] is tackling the same high-level problem. Hence, I think this paper should be discussed and the authors should clarify how their contributions add value beyond existing findings in [1].

[1] Vaezipoor, Pashootan, et al. "Ltl2action: Generalizing ltl instructions for multi-task rl." International Conference on Machine Learning. PMLR, 2021.

**Experimental Designs Or Analyses:**

Apart from the minor question I posed earlier, the experimental design appears sound and valid.

**Methods And Evaluation Criteria:**

The benchmark environments seem appropriate and the authors present experiments on a wide range of environments (including standard MuJoCo environments). The evaluation criterion also makes sense. One thing that is not very clear is how the satisfaction of an output trajectory is calculated. For the evaluation to make sense, the generated trajectories do not only need to satisfy the given specification but also need to be realizable (follow the dynamics in the environment). It is not apparent from reading the paper that the authors are considering realizability in the evaluations.

**Other Comments Or Suggestions:**

N/A

**Other Strengths And Weaknesses:**

- In spite of the similarity with prior work, the idea of using a flow model in this context appears novel and natural. The results look encouraging since the required number of flow steps for a decent performance is small (in the study in the paper), which improves the time taken for planning.
- Clarity can be improved slightly and some things can be made more precise (see questions below).

**Questions For Authors:**

- Why is there a distinction between $\phi_{and}$ and $\phi_{or}$ in page 4? From the definition, it looks like a $\phi_{or}$ formula is also a $\phi_{and}$ formula and vice-versa. Aren't they just equivalent to $\phi_{bool} := \phi_{reach}\ |\ \phi_{bool,1}\lor\phi_{bool,2}\ |\ \phi_{bool,1}\land\phi_{bool,2}$? Then $\phi_{multi}$ can also be written as just $\phi_{bool}\land\phi_{avoid}$.
- In page 4, the description of $\phi_{partial}$ is confusing. It is mentioned that $O_p$ has to be reached after $O_q$ is reached. But $(\lnot O_p)\ \mathcal{U}\ O_q$ is satisfied even if $O_p$ is never reached.
- Is it possible to define an executable policy from the trajectory generated by the model? If so, why not measure the satisfaction of the specification based the trajectory resulting from executing this policy (instead of directly analyzing the generated trajectory which may not exactly follow system dynamics)?

**Relation To Broader Scientific Literature:**

- The presented approach can be considered as a meta reinforcement learning algorithm. STL offers a natural syntax to encode specifications in the context of meta RL.
- Although there has been a lot of recent work on reinforcement learning from temporal logic specifications, there is relatively less research on zero shot execution given a new specification. This paper attempts to tackle this important problem.
- Obtaining good embeddings of code (in this case, STL specifications) is a relevant problem. For instance, it is known that LLMs do not understand the semantics of code very well (even though they can generate good code). It appears that the approach in the paper enables the agent to understand the semantics of STL specifications.

**Theoretical Claims:**

There are no new theoretical claims in the paper.

---

> ### Author Rebuttal · Authors · 2025-04-01
>
> We thank Reviewer 8aYy for the thoughtful feedback and appreciation of our ablation studies and graph-based encoding, which aim to enhance symbolic understanding in decision-making. Below, we address the comments on novelty and trajectory realizability.
>
> **Significance of STL planning**: (same to Reviewer 8mCx) We argue that it is non-trivial to extend the existing LTL2Action [1] to STL, particularly for the key technique “progression” [2] used to update task spec based on assignments. Consider an example (Sec 3.3 in LTL2Action) “First reach R, then reach G”. In LTL, this can be written as $F(R \land F(G))$. Once reach R, following LTL2Action, the LTL is updated to $F(G)$, and the updated reward will encourage reaching G.
>
> STL expresses this in $F_{[ta,tb]} (R \land F_{[tc,td]} G)$ where [ta, tb] is the time range to reach R, and [tc, td] is the relative range that G should be reached after reaching R. Now, the progression is non-trivial - when reaching R, we need to: (1) ensure the event “reach R” happens in [ta, tb] (2) store this event time because the success of “reach G” will depend on the time R was reached. So we need to bookkeep all the “reach R” events as well as their times, and whenever we reach G, we need to iterate all the past “reach R” events to check if the “reach G” event happens after any of it within the [tc,td] range. The complexity is $O(T^2)$, where T is the trace length. The complexity is $O(T^L)$ for L nested temporal layers in STL. Thus, it is not trivial nor efficient to extend LTL2Action to STL.
>
> Because STL doesn’t have automata-like forms as LTL does, it is hard to efficiently augment the state to be Markovian, as mentioned in our paper (page 2, lines 81-83, right column). Thus we use imitation learning to learn diverse STL, and we believe this distinction highlights the novelty and our contribution. That said, we value the reviewer’s observation, and we will cite LTL2Action [1] and the progression paper [2] with explanations in the final version.
>
> **Realizability**: (same to Reviewer GKRz) We appreciate the reviewer’s question regarding dynamics consistency. In all cases, our method generates waypoint trajectories, regardless of whether actions are also predicted during training. These waypoints serve as high-level plans, which can directly get actions from inverse dynamics or be tracked by PD controllers. While we do not enforce dynamics during inference, our focus is on generating STL-compliant high-level paths. Ensuring executability can be done through future extensions such as trajectory refinement or policy warm-starting, as explored in prior works [3], and we will clarify this in the final version.
>
> **Q1: $\phi_{and}$ and $\phi_{or}$**: This is a good observation, and the distinction was intentionally designed: $\phi_{and}$ and $\phi_{or}$ are used to enforce “structured”, canonical forms: $\phi_{and}$ requires all its children nodes to be “disjunctions” (OR-type) while $\phi_{or}$ requires its children to be conjunctions (AND-type). The reviewer's proposed formula will allow both cases. For instance, we allow STL formulas like $A \lor (B \land C)$ but do not allow redundant nesting such as $A \land (B \land C)$, which can be flattened into $A \land B \land C$. This structural design eases the NN training by reducing the variability of data in syntactic forms.
>
> **Q2: $\phi_{partial}$**: Sorry for the confusion. In fact, in our implementation, we additionally add “Eventually Reach $O_p$” to ensure the agent will first reach $O_q$ and then move to $O_p$ (as shown in our appendix, Figures 19, 20). In some other cases (Figure 37 in our appendix for the PointMaze env), we don’t explicitly specify to reach K0 and K1, but from the layout the agent has to reach the keys first. We will make these clear in our final paper.
>
> **Q3: measure in executable policy**: It is possible to define the executable policy for simple cases like Linear and Dubins from the predicted waypoint trajectories, but hard for PointMaze (higher-order ODE for dynamics), AntMaze (higher-order ODE, contact force) or Franka Panda (inverse kinematics to convert workspace trajectories to the configuration space to derive the policy). We can measure satisfaction by executing this calculated policy in some cases, but as mentioned in the **Realizability** section, we focus on planning STL-compliant high-level paths. Ensuring executability can be done through future extensions as explored in [3], and we will clarify it in the final version.
>
> **References:**
> 1. Vaezipoor, Pashootan, et al. "Ltl2action: Generalizing ltl instructions for multi-task rl." ICML 2021.
> 2. Bacchus, Fahiem, and Froduald Kabanza. "Using temporal logics to express search control knowledge for planning." Artificial intelligence 116.1-2 (2000): 123-191.
> 3. Ajay, Anurag, et al. "Is conditional generative modeling all you need for decision-making?." ICLR 2023

---

> > ### Comment · Reviewer_8aYy · 2025-04-05
> >
> > Thanks for the detailed response. Overall I am still leaning towards acceptance. The lack of executable policy remains the biggest weakness but the authors have suggested concrete extensions to their work to obtain executable policies.

---

### Official Review · Reviewer_GKRz · 2025-03-05

**Overall Recommendation:** 4

**Summary:**

This paper provides a flow-matching based approach to generate plans for a diverse range of Signal Temporal Logic (STL) specifications. Consequently, the proposed method can be optimized to be significantly faster than a diffusion-based approach as considered previously by skipping Ordinary Differential Equation (ODE) steps in the Euler sampling process. A Graphical Neural Network (GNN) encodes the STL specification which is then fed into a conditional flow model that predicts a satisfying trajectory given the current state. Experiments on simple differentiable environments as well as more complicated Maze-like non-differentiable environments show the merits of the approach in speed and trajectory performance.

## update after rebuttal

I appreciate the additional experiments (MILP planner) provided by the authors and the clarifications. It would be beneficial for the usability of the approach if the realizability of the trajectories generated was further discussed or evaluated. Regardless, the paper provides an interesting direction for STL-guided planning and I am still in favor of accepting the paper.

**Claims And Evidence:**

The approach considered (to the best of my knowledge) is the first generative model for planning over STL specifications. The specifications considered are diverse and the experiments are shown on a variety of benchmarks ranging from simple linear dynamics to more complex non-differentiable settings like AntMaze.

**Essential References Not Discussed:**

Mixed-integer linear programming (MILP)-based methods (Kurtz & Lin 2022) are mentioned but not evaluated even for the environments with simple dynamics like Linear and Dubins. To this reviewer, an additional baseline for the non-differentiable Maze environments can be assuming simple linear dynamics to generate waypoints using an MILP planner with the STL spec. objectives and obstacles as Avoid predicates. Since an A* planner is used between the proposed waypoints, the actual trajectories should similarly be realizable (if a path is found).

Some relevant work on robotic control for Signal Temporal Logic is missing:

[1] Co-learning Planning and Control Policies Constrained by Differentiable Logic Specifications, Xiong et al, ICRA 2024

[2] Reinforcement Learning of Flexible Policies for Symbolic Instructions with Adjustable Mapping Specifications, Hatanaka et al, RA-L

[3] Synthesis of Temporally-Robust Policies for Signal Temporal Logic Tasks using Reinforcement Learning, Wang et al, ICRA 2024

**Experimental Designs Or Analyses:**

The STL satisfaction rate calculation is a little hard to understand. The highest STL score trajectory out of 1024 sampled outputs is denoted as the final trajectory. The satisfaction rate is quantified as the ratio of final trajectories that satisfy the STL specification out of 256 specs from the validation set.

Additionally, it is unclear how the trajectories are determined to be realistic without using the environment dynamics directly. For the differentiable environments (Linear, Dubins, Franka Panda), it would be helpful to clarify how the trajectories are deemed to achievable by a given controller. Unlike the proposed method (TeloGraD/F), the **Grad** baseline differentiates through the environment’s transition function and yields trajectories that respect the system dynamics.

Error bars representing standard deviation or spread are not present in any of the results considered.

**Methods And Evaluation Criteria:**

I am mostly in agreement with the experiment setting and appreciate the thoroughness in finding a diverse range of STL specifications and goals to support the claims of generality.

It appears that the robustness score ($\rho$) of the STL specification ($\phi$) is not used during evaluation time and the GNN encoding of $\phi$ alone is used in the conditional flow model.  Using the robustness score as part of the guidance (akin to LTLDoG) could in fact make TeLoGraD a stronger technique.

For the non-differentiable Maze environments, the time to satisfy the specification or length of the trajectories considered should be helpful to distinguish between the plan qualities of the various methods.

**Other Comments Or Suggestions:**

- p6L310 : referencing the graphs mentioned (Fig. 5) would help clarify if they are different from Fig. 4.
- p12L639 : The equation referenced is wrong (should be Eq. 7)

**Other Strengths And Weaknesses:**

The ability to generate satisfying plans quickly is especially interesting and begs the question whether slower inference diffusion models, that could be more powerful and diverse, are necessary for most environments.

**Questions For Authors:**

1. Is the robustness score used at all during evaluation (and not only during generating the demonstration data)? On a related note , how is this gradient based planner used for demonstration data implemented (for the differentiable environments)? Does it differentiate through the dynamics? How many time steps are the given trajectories for Linear and Dubins?
2. It appears that the **Grad** baseline significantly outperforms **TeloGraD/F** in Linear environment. Is this because, unlike **Grad,** the proposed method does not use environment dynamics which can help significantly when the environment is simple (like in Linear)?
3. How are the generated trajectories verified to be consistent with environment dynamics?
4. Is the same methodology to calculate satisfaction rates carried out for all baselines? (the same 256 specs from the validation set followed by sampling 1024 trajectories to pick 256 final trajectories). If this understanding is incorrect, an explanation would be appreciated.
5. Can there be comparisons provided with MILP based methods for the differentiable environments?  If this is not an appropriate baseline, why?

**Relation To Broader Scientific Literature:**

The study and modern dataset for diverse STL specifications along with a release for the GNN-based approach to encode the objectives is relevant and will be appreciated by the community. Diffusion based methods using STL as guidance have been considered previously but not in the general planning setting (except LTLDoG to the best of my knowledge).

**Theoretical Claims:**

No novel theoretical advances were discussed.

---

> ### Author Rebuttal · Authors · 2025-04-01
>
> We appreciate Reviewer GKRz's thorough review and positive evaluations. We are glad to see that our contributions (first STL generative model; diverse setups; fast inference) have been recognized. We conducted new experiments as requested. Below are our responses to the concerns and questions.
>
> **(New experiment) MILP baseline**: We use the official codebase from [1] to run MILP baseline on Linear and PointMaze benchmarks. Dubins and AntMaze are omitted due to their nonlinear dynamics and similarity to Linear and PointMaze, respectively. We omit Franka Panda due to its nonlinear constraints from the configuration space. In Linear, we approximate the circular goal/obstacle regions using polygons. We run MILP under various timeouts (“second”):
>
> |Method | STL satisfaction | Runtime (s)|
> |-|-|-|
> |MILP (5) | 0.062 | 5.147|
> |MILP (10) | 0.203 | 18.682|
> |MILP (30) | 0.328 | 27.178|
> |MILP (45) | 0.375 | 37.131|
> |MILP (60) | 0.375 | 47.508|
> |MILP (90) | 0.383 | 68.913|
> |**TeLoGraF(Fast)** | **0.477** |**0.174**|
>
> The result shows that TeLoGraF(Fast) outperforms MILP (90) in STL satisfaction, with 400X speed-up (MILP runtime is reasonable compared to TABLE I in paper [1]). MILP is slow as it generates thousands of binary variables in our long-horizon tasks (horizon T=64). We further ran MILP on “PointMaze” (horizon T=512) and it only achieves 0.02 STL satisfaction with average runtime 329.785 seconds, highlighting the efficiency and scalability of our method for long-horizon STL planning.
>
> **(New experiment) Evaluation with varied random seeds**: We redo the experiment in Figure 4 with three random seeds to generate error bars - for learning-based methods, we train and evaluate the NN from different seeds; for non-learning methods, we directly evaluate them with different seeds. Due to time constraints, current results are just for the first three benchmarks, and we will complete this figure by the end of the rebuttal phase. The updated Figure 4 is at https://postimg.cc/XX4LtW62 - the statistical trends remain consistent with those reported in our submission.
>
> **Q1: implementation details**: For each STL spec in the validation set, the robustness score is only used in evaluation to select the best quality trajectory (out of 1024 sampled trajectories). We have 256 STL specs in the validation set. For the gradient-based planner, we use “backprop-through-time” (BPTT) [2] to optimize trajectories thus, it differentiates through the dynamics. The timesteps are 64 for Linear, Dubins and Franka Panda, and 512 for PointMaze and AntMaze.
>
> **Q2: Grad outperforms TeLoGraD/F**: This is an interesting finding, and we agree with the reviewer’s insight. Another factor is that “Linear” has simple, fully actuated dynamics, making it well-suited for gradient-based optimization. In contrast, “Dubins” involves second-order dynamics (acceleration -> speed -> position) with limited control, making them more “dragged” and underactuated and hindering gradient-based planners from efficiently finding solutions.
>
> **Q3: Realistic trajectories**: We appreciate the reviewer’s question regarding dynamics consistency. In all cases, our method generates waypoint trajectories, regardless of whether actions are also predicted during training. These waypoints serve as high-level plans, which can directly get actions from inverse dynamics or be tracked by PD controllers. While we do not enforce dynamics during inference, our focus is on generating STL-compliant high-level paths. Ensuring executability can be done through future extensions such as trajectory refinement or policy warm-starting, as explored in prior works [3], and we will clarify this in the final version.
>
> **Q4: STL metrics**: The STL satisfaction rate is evaluated over 256 STL specs on the validation set (unless otherwise specified, like in Figure 5). For each STL spec, we use our method or other baselines to generate 1024 trajectories and pick the one with the highest STL score as the final trajectory. So these 256 trajectories (one for each STL spec) will be marked as 1 (satisfaction) or 0 (STL violation). And the STL satisfaction rate is computed across these binary outcomes.
>
> **Q5: MILP baseline**: Thanks for the suggestion; we have conducted the experiment showing “TeLoGraF (Fast)”’s advantage over MILP for long-horizon STL tasks.
>
> **Missing STL references**: Thanks. We will cite those accordingly.
>
> **Other suggestions**: Thanks for pointing out the potential improvement in P6, L310, and the typo in P12, L639. We will modify the paper in the final version.
>
> **References:**
> 1. Kurtz, Vincent, and Hai Lin. "Mixed-integer programming for signal temporal logic with fewer binary variables." IEEE Control Systems Letters 6 (2022): 2635-2640.
> 2. Leung, Karen, and Marco Pavone. "Semi-supervised trajectory-feedback controller synthesis for signal temporal logic specifications." ACC 2022.
> 3. Ajay, Anurag, et al. "Is conditional generative modeling all you need for decision-making?." ICLR 2023.

---

### Official Review · Reviewer_8mCx · 2025-03-14

**Overall Recommendation:** 2

**Summary:**

This paper proposes to use learning-based methods to generate planning trajectories for STL specifications. The authors introduce some STL templates and then present a GNN to encode the STLs into feature representations. These features are then fed into flow matching as the conditioning factor for end-to-end learning of valid trajectories.

**Claims And Evidence:**

Partially. The authors claim that most previous work do not consider diverse STL or neural-network encoder, while there are existing work that can achieve this for a similar temporal logic -- linear temporal logic (LTL). Although their semantics are not entirely same, many template and encoder designs can be shared and re-utilized with minor modifications.

Vaezipoor P, Li AC, Icarte RA, Mcilraith SA. Ltl2action: Generalizing ltl instructions for multi-task rl. InInternational Conference on Machine Learning 2021 Jul 1 (pp. 10497-10508). PMLR.

**Essential References Not Discussed:**

Please see Claims And Evidence part for a previous work LTL2Action.

**Experimental Designs Or Analyses:**

Experiments are reasonable.

**Methods And Evaluation Criteria:**

The proposed methodology is reasonable. The proposed dataset can be used to test the proposed algorithm. However, it is unclear how difficult or laborious it is for a new environment in real applications.

**Other Comments Or Suggestions:**

No.

**Other Strengths And Weaknesses:**

The contribution of STL templates and GNN encoder is incremental compared to LTL2Action as the extension of these for LTL to STL does not face significant challenges. Therefore, the technical contribution is limited.

The proposed method requires paired demonstrations for training the conditional generative model. If I am understanding correctly, this would require first collecting a set of demonstrations for a new environment in application before the method can be trained. In real physical world this might still be very laborious.

**Questions For Authors:**

No.

**Relation To Broader Scientific Literature:**

Will be interesting to the venue and researchers focusing on STL and neural-symbolic learning.

**Theoretical Claims:**

No theoretical proofs.

---

> ### Author Rebuttal · Authors · 2025-04-01
>
> We thank the reviewer for the thoughtful reviews. Below are our responses to the concerns.
>
> **Extend LTL work to STL**: We argue that it is non-trivial to extend the existing LTL2Action [1] to STL, particularly for the key technique “progression” [2] used to update task spec based on assignments. Consider an example (Sec 3.3 in LTL2Action) “First reach R, then reach G”. In LTL, this can be written as $F(R \land F(G))$. Once it reaches R, following LTL2Action, the LTL is updated to $F(G)$, and the updated reward will encourage reaching G.
>
> STL expresses this in $F_{[ta,tb]} (R \land F_{[tc,td]} G)$ where [ta, tb] is the time range to reach R, and [tc, td] is the relative range that G should be reached after reaching R. Now, the progression is non-trivial - when reaching R, we need to: (1) ensure the event “reach R” happens in [ta, tb] (2) store this event time because the success of “reach G” will depend on the time R was reached. So we need to bookkeep all the “reach R” events as well as their times, and whenever we reach G, we need to iterate all the past “reach R” events to check if the “reach G” event happens after any of it within the [tc,td] range. The complexity is $O(T^2)$, where T is the trace length. The complexity is $O(T^L)$ for L nested temporal layers in STL. Thus, it is not trivial nor efficient to extend LTL2Action to STL.
>
> Because STL doesn’t have automata-like forms as LTL does, it is hard to efficiently augment the state to be Markovian, as mentioned in our paper (page 2, lines 81-83, right column). Thus we use imitation learning to learn diverse STL, and we believe this distinction highlights the novelty and our contribution. That said, we value the reviewer’s observation, and we will cite LTL2Action [1] and the progression paper [2] with explanations in the final version.
>
> **Data collection labor in real-world**: We understand reviewer’s concern regarding paired expert data. However, we note that learning from demonstrations for robots is a widely adopted paradigm in both simulation and real-world settings. In cases where no efficient solver exists, it is common to collect demonstrations from scripted policies, human operators, or off-the-shelf planners. STL belongs to this category, as mentioned in our paper (page 1, L25-L36, right column). We view our work as a first step toward enabling STL-conditioned policy learning from demonstrations—an area that is currently underexplored and, to our knowledge, lacks dedicated prior work.
>
> Although we consider the paired data in our work, in the future, demonstrations do not need to be paired with STL. Recent works have shown STL can be inferred from offline demonstrations [3,4,5,6] or translated from natural language [7,8,9]. With these techniques and the increasing availability of open and modular data collection pipelines (e.g., UMI [10]), we argue that the reliance on demonstrations should not be viewed as a major limitation.
>
> **Contribution:** We respectfully disagree that our contribution is incremental. As acknowledged by the other reviewers, our work introduces a novel conditional flow model for general STL planning with solid experiments. Reviewer GKRz appreciated the diverse STL coverage and fast generation of satisfying plans, and Reviewer 8aYy highlighted that our model effectively captures STL semantics with efficient inference. During rebuttal, we also conduct extra experiments, including evaluations across varied random seeds and compare with an MILP baseline. These perspectives support the relevance of our contributions and their potential impact on the STL planning community.
>
> We hope this addresses the concerns, and we welcome further discussions.
>
> **References:**
> 1. Vaezipoor, Pashootan, et al. "Ltl2action: Generalizing ltl instructions for multi-task rl." International Conference on Machine Learning. PMLR, 2021
> 2. Bacchus, Fahiem, and Froduald Kabanza. "Using temporal logics to express search control knowledge for planning." Artificial intelligence 116.1-2 (2000): 123-191
> 3. Liu, Wenliang, et al. "Interpretable generative adversarial imitation learning." arXiv 2024
> 4. Leung, Karen, and Marco Pavone. "Semi-supervised trajectory-feedback controller synthesis for signal temporal logic specifications." ACC 2022
> 5. Meng, Yue, and Chuchu Fan. "Diverse controllable diffusion policy with signal temporal logic." RA-L 2024
> 6. Vazquez-Chanlatte, Marcell, et al. "Learning task specifications from demonstrations." NeurIPS 2018
> 7. Shah, Ankit, et al. "Bayesian inference of temporal task specifications from demonstrations." NeurIPS 2018
> 8. Cosler, Matthias, et al. "nl2spec: Interactively translating unstructured natural language to temporal logics with large language models." CAV 2023
> 9. He, Jie, et al. "Deepstl: from english requirements to signal temporal logic." International Conference on Software Engineering. 2022
> 10. Chi, Cheng, et al. "Universal manipulation interface: In-the-wild robot teaching without in-the-wild robots." RSS 2024

---

### Decision · Program_Chairs · 2025-05-01

**Decision:**

Accept (poster)

**Comment:**

The reviewers acknowledged that the paper presents a novel approach, TeLoGraF, which combines graph neural networks and flow-matching to solve general signal temporal logic (STL) planning tasks. The authors successfully demonstrated the method's effectiveness across a diverse set of simulation environments, significantly improving STL satisfaction rates and offering substantial runtime efficiency compared to classical algorithms. Although concerns were raised regarding incremental contributions relative to similar LTL-based methods, the authors' rebuttal convincingly addressed these points, clarifying the complexity and uniqueness of extending their approach from LTL to STL. The reviewers also highlighted the paper's limitations, such as reliance on pre-collected demonstrations and lack of executable policies, but recognized the practical pathways proposed for future work. Overall, given the robustness of the method, convincing experimental validation, and effective author responses, this submission represents a solid contribution and warrants acceptance.